# Uncovering Untapped Potential in Sample-Efficient World Model Agents

## Abstract

World model (WM) agents enable sample-efficient reinforcement learning by learning policies entirely from simulated experience. However, existing token-based world models (TBWMs) are limited to visual inputs and discrete actions, restricting their adoption and applicability. Moreover, although both intrinsic motivation and prioritized WM replay have shown promise in improving WM performance and generalization, they remain underexplored in this setting, particularly in combination. We introduce Simulus, a highly modular TBWM agent that integrates (1) a modular multi-modality tokenization framework, (2) intrinsic motivation, (3) prioritized WM replay, and (4) regression-as-classification for reward and return prediction. Simulus achieves state-of-the-art sample efficiency for planning-free WMs across three diverse benchmarks. Ablation studies reveal the individual contribution of each component while highlighting their synergy. Our code and model weights are publicly available at `https://anonymous.4open.science/r/Simulus-FBF5`.

## 1  Introduction

Sample efficiency refers to the ability of a reinforcement learning (RL) algorithm to learn effective policies using as few environment interactions as possible. In many real-world domains such as robotics, autonomous driving, and healthcare, this is particularly critical, as interactions are costly, slow, or constrained. World model agents, methods that learn control entirely from simulated experience generated by a learned dynamics model, have emerged as a promising approach to improving sample efficiency [17, 20, 37, 57].

Here, we focus on token-based world models (TBWMs) [37, 38, 11], sample-efficient RL methods that learn the dynamics entirely within a learned discrete token space, where each observation comprises a sequence of tokens. Evidently, most large scale world models [1, 12, 14] operate on multi-token observations, suggesting that such representations are advantageous at scale. TBWMs offer a clear modular design, separating the optimization of its representation, dynamics, and control models. As modular systems, TBWMs are easier to scale, develop, study, and deploy, as individual modules can be treated independently without interfering and are easier to master through divide and conquer. In addition, such separation leads to simpler optimization objectives and avoids interference between them (Appendix B.1).

However, existing TBWMs are restricted to image observations and discrete actions, such as Atari games, limiting both their adoption and broader applicability, as their effectiveness in diverse environments and modalities remains unclear. While multi-modality tokenization approaches exist for large-scale offline settings [43, 47, 33, 34], these methods rely on large vocabularies (e.g., 33K tokens) which are inefficient for online, data-limited regimes. Whether substantially smaller vocabularies can preserve competitive precision and performance is still an open question, leaving these approaches underexplored for sample-efficient RL.

Submitted to 39th Conference on Neural Information Processing Systems (NeurIPS 2025). Do not distribute.

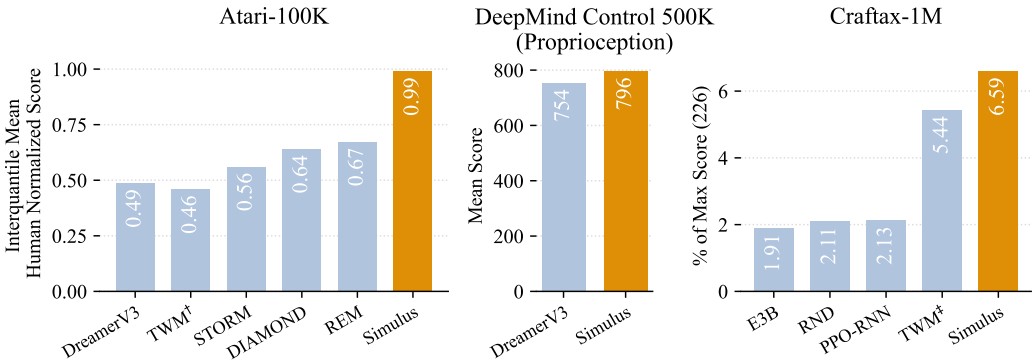

Figure 1: Results overview. Simulus exhibits state-of-the-art sample-efficiency performance for planning-free methods across all three benchmarks. † [44], ‡ [13].

Furthermore, despite compelling results [46, 49, 30], intrinsic motivation and prioritized *world model* replay remain underexplored in sample-efficient world model agents [19, 20, 60, 11, 3], particularly in combination. We conjecture that intrinsic motivation is underused as it may steer the agent toward task-irrelevant regions, potentially wasting limited interaction budget. Prioritized replay [30], while promising, lacks robust empirical support and proved challenging to tune in our experiments.

To address these limitations, we propose Simulus, a modular world model that extends a recent TBWM method [11] by integrating several powerful advances from the literature: (1) a modular multi-modality tokenization framework for handling arbitrary combinations of observation and action modalities, (2) intrinsic motivation for epistemic uncertainty reduction [50, 49], (3) prioritized world model replay [30], and (4) regression-as-classification (RaC) for reward and return prediction [16, 20].

To evaluate the impact of the proposed components, we conducted extensive empirical evaluations across three diverse benchmarks, ranging from the visual Atari 100K [28], to the continuous proprioception tasks of the DeepMind Control Suite [53], to Craftax [35], which combines symbolic 2D grid maps with continuous state features. There, Simulus achieves state-of-the-art sample-efficiency for planning-free world models. Ablation studies further show the contribution and effectiveness of each component.

Summary of contributions:

- Through extensive empirical evaluations across three diverse benchmarks, we show that intrinsic motivation, prioritized replay, and RaC significantly improve performance in the sample-efficiency setting for world model agents, particularly when combined.

- Our results demonstrate the effectiveness of the multi-modality tokenization approach, as Simulus achieves state-of-the-art planning-free performance across benchmarks, establishing TBWMs as widely-applicable methods.

- We propose Simulus, a versatile TBWM agent that follows a highly modular design, offering a solid foundation for future developments. To support future research and adoption of TBWMs, we open-source our code and weights.

## 2 Method

**Notations** We consider the Partially Observable Markov Decision Process (POMDP) setting. However, since in practice the agent has no knowledge about the hidden state space, consider the following state-agnostic formulation. Let $\Omega, \mathcal{A}$ be the sets of observations and actions, respectively. At every step $t$, the agent observes $\mathbf{o}_t \in \Omega$ and picks an action $\mathbf{a}_t \in \mathcal{A}$. From the agent's perspective, the environment evolves according to $\mathbf{o}_{t+1}, r_t, d_t \sim p(\mathbf{o}_{t+1}, r_r, d_t | \mathbf{o}_{\leq t}, \mathbf{a}_{\leq t})$, where $r_t, d_t$ are the observed reward and termination signals, respectively. The process repeats until a positive termination signal $d_t \in \{0, 1\}$ is obtained. The agent's objective is to maximize its expected return $\mathbb{E}[\sum_{t=0}^{\infty} \gamma^t r_{t+1}]$ where $\gamma \in [0, 1]$ is a discount factor.

For multi-modal observations, let $\mathbf{o}_t = \{\mathbf{o}_t^{(i)}\}_{i=1}^{|\kappa|}$ where $\kappa$ is the set of environment modalities and $\mathbf{o}_t^{(i)}$ denotes the features of modality $\kappa_i$.

**Overview**   Simulus builds on REM [11]. The agent comprises a representation module $\mathcal{V}$, a world model $\mathcal{M}$, a controller $\mathcal{C}$, and a replay buffer. To facilitate a modular design, following REM, each module is optimized separately. The training process of the agent involves a repeated cycle of four steps: data collection, representation learning ($\mathcal{V}$), world model learning ($\mathcal{M}$), and control learning in imagination ($\mathcal{C}$).

## 2.1 The Representation Module $\mathcal{V}$

$\mathcal{V}$ is responsible for encoding and decoding raw observations and actions. It is a modular tokenization system with encoder-decoder pairs for different input modalities. Encoders produce fixed-length token sequences, creating a common interface that enables combining tokens from various sources into a unified representation. After embedding, these token sequences are concatenated into a single representation, as described in Section 2.2. Note that encoder-decoder pairs need not be learning-based methods; however, when learned, they are optimized independently. This design enables $\mathcal{V}$ to deal with any combination of input modalities, provided the respective encoder-decoder pairs.

**Tokenization**   $\mathcal{V}$ transforms raw observations $\mathbf{o}$ to sets of fixed-length integer token sequences $\mathbf{z} = \{\mathbf{z}^{(i)}\}_{i=1}^{|\kappa|}$ by applying the encoder of each modality $\mathbf{z}^{(i)} = \mathrm{enc}_i(\mathbf{o}^{(i)})$. Actions $\mathbf{a}$ are tokenized using the encoder-decoder pair of the related modality to produce $\mathbf{z}^{\mathrm{a}}$. The respective decoders reconstruct observations from their tokens: $\hat{\mathbf{o}}^{(i)} = \mathrm{dec}_i(\mathbf{z}^{(i)})$.

Simulus natively supports four modalities: images, continuous vectors, categorical variables, and image-like multi-channel grids of categorical variables, referred to as "2D categoricals". More formally, 2D categoricals are elements of $([k_1] \times [k_2] \times \ldots \times [k_C])^{m \times n}$ where $k_1, \ldots, k_C$ are per channel vocabulary sizes, $C$ is the number of channels, $m, n$ are spatial dimensions, and $[k] = \{1, \ldots, k\}$.

Following REM, we use a VQ-VAE [15, 55] for image observations. For the tokenization of continuous vectors, each feature is quantized to produce a token, as in [43]. However, to improve learning efficiency, we reduce the vocabulary size from 1024 to 125, and modify the quantization levels for optimal coverage (Appendix A.2.2). Unbounded vectors are first transformed using the symlog function [20], defined as $\mathrm{symlog}(x) = \mathrm{sign}(x)\ln(1 + |x|)$, which compresses the magnitude of large absolute values. Lastly, while no special tokenization is required for categorical inputs, 2D categoricals are flattened along the spatial dimensions to form a sequence of categorical vectors. The embedding of each token vector is obtained by averaging the embeddings of its entries.

## 2.2 The World Model $\mathcal{M}$

The purpose of $\mathcal{M}$ is to learn a model of the environment's dynamics. Concretely, given trajectory segments $\tau_t = \mathbf{z}_1, \mathbf{z}_1^{\mathrm{a}}, \ldots, \mathbf{z}_t, \mathbf{z}_t^{\mathrm{a}}$ in token representation, $\mathcal{M}$ models the distributions of the next observation and termination signal, and the expected reward:

$$\text{Transition:} \quad p_\theta(\hat{\mathbf{z}}_{t+1}|\tau_t), \tag{1}$$

$$\text{Reward:} \quad \hat{r}_t = \hat{r}_\theta(\tau_t), \tag{2}$$

$$\text{Termination:} \quad p_\theta(\hat{d}_t|\tau_t), \tag{3}$$

where $\theta$ is the parameters vector of $\mathcal{M}$ and $\hat{r}_\theta(\tau_t)$ is an estimator of $\mathbb{E}_{r_t \sim p(r_t|\tau_t)}[r_t]$.

**Architecture**   $\mathcal{M}$ comprises a sequence model $f_\theta$ and multiple heads for the prediction of tokens of different observation modalities, rewards, termination signals, and for the estimation of model uncertainty. Concretely, $f_\theta$ is a retentive network (RetNet) [51] augmented with a parallel observation prediction (POP) [11] mechanism. All heads are implemented as multilayer perceptrons (MLP) with a single hidden layer. We defer the details about these architectures to Appendix A.3.

**Embedding**   $\mathcal{M}$ translates token trajectories $\tau$ into sequences of $d$-dimensional embeddings $\mathbf{X}$ using a set of embedding (look-up) tables. By design, each modality is associated with a separate

table. In cases where an embedding table is not provided by the appropriate encoder-decoder pair, $\mathcal{M}$ and $\mathcal{C}$ learn dedicated tables separately and independently. As embeddings sequences are composed hierarchically, we use the following hierarchical notation:

$$\text{Observation-action block:} \quad \mathbf{X}_t = (\mathbf{X}_t^{\mathrm{o}}, \mathbf{X}_t^{\mathrm{a}}),$$

$$\text{Observation block:} \quad \mathbf{X}_t^{\mathrm{o}} = (\mathbf{X}_t^{(1)}, \ldots, \mathbf{X}_t^{(|\kappa|)}),$$

where $K_i$ denotes the number of embedding vectors in $\mathbf{X}_t^{(i)}$. Similarly, $K = \sum_{i=1}^{|\kappa|} K_i$. To combine latents of each $\mathbf{z}_t$, $\mathcal{V}$ concatenates their token sequences along the temporal axis based on a predefined modality order. We defer the full details on the embedding process to Appendix A.3.

**Sequence Modeling** Given a sequence of observation-action blocks $\mathbf{X} = \mathbf{X}_1, \ldots, \mathbf{X}_t$, the matching outputs $\mathbf{Y}_1, \ldots, \mathbf{Y}_t$ are computed auto-regressively as follows:

$$(\mathbf{S}_t, \mathbf{Y}_t) = f_\theta(\mathbf{S}_{t-1}, \mathbf{X}_t),$$

where $\mathbf{S}_t$ is a recurrent state that summarizes $\mathbf{X}_{\leq t}$ and $\mathbf{S}_0 = 0$. However, the output $\mathbf{Y}_{t+1}^{\mathrm{u}}$, from which $\hat{\mathbf{z}}_{t+1}$ is predicted, is computed using the POP mechanism via another call as

$$(\cdot, \mathbf{Y}_{t+1}^{\mathrm{u}}) = f_\theta(\mathbf{S}_t, \mathbf{X}^{\mathrm{u}}),$$

where $\mathbf{X}^{\mathrm{u}} \in \mathbb{R}^{K \times d}$ is a learned embedding sequence. Intuitively, $\mathbf{X}^{\mathrm{u}}$ acts as a learned prior, enabling the parallel generation of multiple tokens into the future.

To model $p_\theta(\hat{\mathbf{z}}_{t+1} | \mathbf{Y}_{t+1}^{\mathrm{u}})$, the distributions $p_\theta(\hat{z} | \mathbf{y})$ of each token $\hat{z}$ of $\hat{\mathbf{z}}_{t+1}^{(i)}$ of each modality $\kappa_i$ are modeled using modality-specific prediction heads implemented as MLPs with a single hidden layer and an output size equal to the vocabulary size of $\mathrm{enc}_i$ (Figure 2b). For 2D categoricals, $C$ heads are used to predict the $C$ tokens from each $\mathbf{y}$.

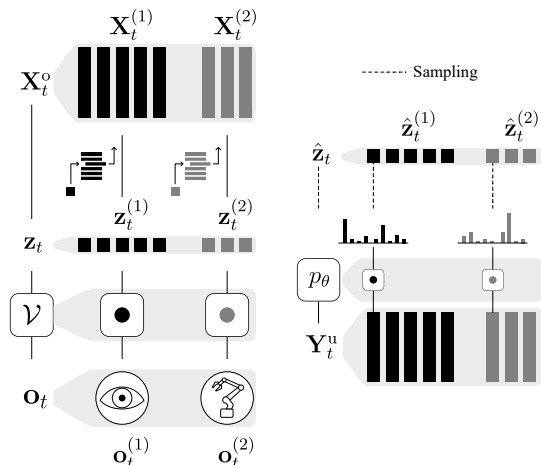

(a) Observation tokenization and embedding.

(b) Prediction of observation tokens.

Figure 2: An illustration of the independent processing of modalities for an observation with two modalities.

Similarly, rewards and termination signals are predicted by additional prediction heads as $\hat{r}_t = \hat{r}_\theta(\mathbf{y})$, $\hat{d}_t \sim p_\theta(\hat{d}_t | \mathbf{y})$, slightly abusing notations, where $\mathbf{y}$ is the last vector of $\mathbf{Y}_{t+1}^{\mathrm{u}}$. An illustration is provided in Figure 3.

**Reducing Epistemic Uncertainty via Intrinsic Motivation** The world model $\mathcal{M}$ serves as the cornerstone of the entire system. Any controller operating within a world model framework can only perform as well as the underlying world model allows, making its quality a fundamental limiting factor. In deep learning methods, the model's performance depends heavily on the quality of its training data. Accurate dynamics modeling requires comprehensive data collection that captures the full spectrum of possible environmental behaviors. This presents a particular challenge in online RL, where the controller must systematically and efficiently explore its environment. Success depends on intelligently guiding the controller toward unexplored or undersampled regions of the dynamics space. An effective approach to this challenge involves estimating the world model's epistemic uncertainty and directing the controller to gather data from regions where this uncertainty is highest [46, 50, 49].

Our approach estimates epistemic uncertainty using an ensemble of $N_{\mathrm{ens}} = 4$ next observation prediction heads $\{p_{\phi_i}(\hat{\mathbf{z}} | \mathrm{sg}(\mathbf{Y}^{\mathrm{u}}))\}_{i=1}^{N_{\mathrm{ens}}}$ with parameters $\{\phi_i\}_{i=1}^{N_{\mathrm{ens}}}$ [49, 31] where $\mathrm{sg}(\cdot)$ is the stop gradient operator. To quantify disagreement between the ensemble's distributions, we employ the Jensen-Shannon divergence (JSD) [50]. For probability distributions $P_1, \ldots, P_n$, the JSD is defined as:

$$\mathrm{JSD}(P_1, \ldots, P_n) = \mathcal{H}\left(\frac{1}{n} \sum_{i=1}^{n} P_i\right) - \frac{1}{n} \sum_{i=1}^{n} \mathcal{H}(P_i),$$

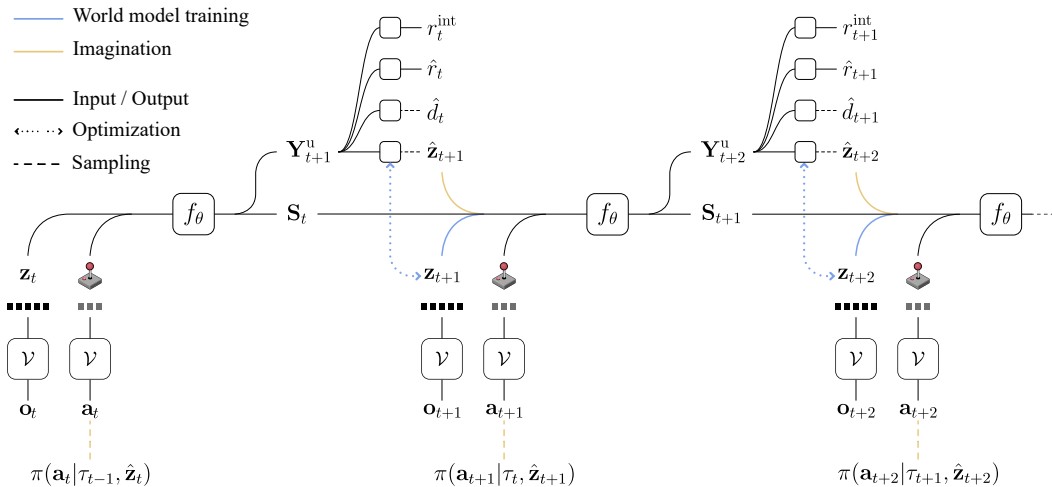

Figure 3: World model training and imagination. To maintain visual clarity, we omitted token embedding details, as well as optimization details of rewards and termination signals.

where $\mathcal{H}(\cdot)$ denotes the Shannon entropy. Since observations comprise multiple tokens, we average the per-token JSD values to obtain a single uncertainty measure $\delta_t$. Training data is divided equally among ensemble members, with each predictor processing a distinct subset of each batch. Despite the ensemble approach, our implementation maintains computational efficiency, with negligible additional overhead in practice.

To guide $\mathcal{C}$ towards regions of high epistemic uncertainty, $\mathcal{M}$ provides $\mathcal{C}$ with additional intrinsic rewards $r_t^{\text{int}} = \delta_t$ during imagination. Here, the reward provided by $\mathcal{M}$ at each step $t$ is given by

$$\bar{r}_t = w^{\text{int}} r_t^{\text{int}} + w^{\text{ext}} \hat{r}_t,$$

where $w^{\text{int}}, w^{\text{ext}} \in \mathbb{R}$ are hyperparameters that control the scale of each reward type. Optimizing the controller in imagination allows it to reach areas of high model uncertainty without additional real-environment interaction.

**Prioritized Replay**    Recent work has demonstrated that prioritizing replay buffer sampling during world model training could lead to significant performance gains in intrinsically motivated agents [30]. While their approach showed promise, it required extensive hyperparameter tuning in practice. We propose a simpler, more robust prioritization scheme for world model training.

Here, the replay buffer maintains a world model loss value for each stored example, with newly added examples assigned a high initial loss value $\nu_0$. During $\mathcal{M}$'s training, we sample each batch using a mixture of uniform and prioritized sampling, controlled by a single parameter $\alpha \in [0, 1]$ that determines the fraction of prioritized samples. For the prioritized portion, we sample examples proportional to their softmax-transformed losses $p_i = \text{softmax}(\mathcal{L})_i$. The loss values are updated after each world model optimization step using the examples' current batch losses.

**Training**    We use the cross-entropy loss for the optimization of all components of $\mathcal{M}$. Specifically, for each $t$, the loss of $p_\theta(\hat{\mathbf{z}}_t | \mathbf{Y}_t^{\text{u}})$ is obtained by averaging the cross-entropy losses of its individual tokens. The same loss is used for each ensemble member $p_{\phi_i}(\hat{\mathbf{z}}_t | \text{sg}(\mathbf{Y}_t^{\text{u}}))$. The optimization and design of the reward predictor is similar to that of the critic, as described in Section 2.3. A formal description of the optimization objective can be found in Appendix A.3.1.

### 2.3   The Controller $\mathcal{C}$

$\mathcal{C}$ is an extended version of the actor-critic of REM [11] that supports additional observation and action spaces and implements regression-as-classification for return predictions.

**Architecture**    At the core of $\mathcal{C}$'s architecture, parameterized by $\psi$, is an LSTM [26] sequence model. At each step $t$, upon observing $\mathbf{z}_t$, a set of modality-specific encoders map each modality tokens $\mathbf{z}_t^{(i)}$ to

a latent vector $\mathbf{x}^{(i)}$, where we abuse our notation $\mathbf{x}$ as the context of the discussion is limited to $\mathcal{C}$. The latents are then fused by a fully-connected network to obtain a single vector $\mathbf{x} = g_\psi(\mathbf{x}^{(1)}, \ldots, \mathbf{x}^{(|\kappa|)})$. $\mathbf{x}_t \in \mathbb{R}^{d_\mathcal{C}}$ is processed by $\mathcal{C}$'s sequence model to produce $\mathbf{h}_t, \mathbf{c}_t = \text{LSTM}(\mathbf{x}_t, \mathbf{h}_{t-1}, \mathbf{c}_{t-1}; \psi)$ where $\mathbf{h}_t, \mathbf{c}_t$ are the LSTM's hidden and cell states, respectively. Lastly, two linear output layers produce the logits from which the actor and critic outputs $\pi(\mathbf{a}_t|\mathbf{h}_t), \hat{V}^\pi(\mathbf{h}_t)$ are derived. For continuous action spaces, the actor uses a categorical distribution over a uniformly spaced discrete subset of $[-1, 1]$. We defer the full details about the encoding process to Appendix A.4.

**Regression as Classification for Reward and Return Prediction**   Robustly handling unbounded reward signals has long been challenging as they can vary dramatically in both magnitude and frequency. Hafner et al. [20] addressed this challenge by using a classification network that predicts the weights of exponentially spaced bins and employed a two-hot loss for the network's optimization. Farebrother et al. [16] studied the use of cross-entropy loss in place of the traditional mean squared error loss for value-based deep RL methods. In their work, the HL-Gauss method was shown to significantly outperform the two-hot loss method. Building on these developments, we adopt the classification network with exponential bins from [20], and apply the HL-Gauss method for its optimization. Concretely, the critic's value estimates are predicted using a linear output layer parameterized by $\mathbf{W} \in \mathbb{R}^{m \times d_\mathcal{C}}$ with $m = 128$ outputs corresponding to $m$ uniform bins defined by $m + 1$ endpoints $\mathbf{b} = (b_0, \ldots, b_m)$. The predicted value is given by

$$\hat{y} = \text{symexp}\left(\text{softmax}(\mathbf{Wh})^\top \hat{\mathbf{b}}\right)$$

where $\text{symexp}(x) = \text{sign}(x)(\exp(|x|) - 1)$ is the inverse of the symlog function and $\hat{\mathbf{b}} = \left(\frac{b_1+b_0}{2}, \ldots, \frac{b_m+b_{m-1}}{2}\right)$ are the bin centers. Given the true target $y \in \mathbb{R}$, the HL-Gauss loss is given by

$$\mathcal{L}_{\text{HL-Gauss}}(\mathbf{W}, \mathbf{h}, y) = \tilde{\mathbf{y}}^\top \log \text{softmax}(\mathbf{Wh})$$

where $\tilde{y}_i = \Phi\left(\frac{b_i - \text{symlog}(y)}{\sigma}\right) - \Phi\left(\frac{b_{i-1} - \text{symlog}(y)}{\sigma}\right)$, $\Phi$ is the cumulative density function of the standard normal distribution and $\sigma$ is a standard deviation hyperparameter that controls the amount of label smoothing.

**Training in Imagination**   $\mathcal{C}$ is trained entirely from simulated experience generated through interaction with $\mathcal{M}$. Specifically, $\mathcal{M}$ and $\mathcal{C}$ are initialized with a short trajectory segment sampled uniformly from the replay buffer and interact for $H$ steps. An illustration of this process is given in Figure 3 (orange path). $\lambda$-returns are computed for each generated trajectory segment and are used as targets for critic learning. For policy learning, a REINFORCE [52] objective is used, with a $\hat{V}^\pi$ baseline for variance reduction. See Appendix A.4.2 for further details.

# 3   Experiments

To evaluate sample efficiency, we used benchmarks that measure performance within a fixed, limited environment interaction budget. These benchmarks were also selected to address key research questions: (1) whether the proposed multi-modality approach is effective—both in continuous control settings and in handling multi-modal observations; and (2) whether the integrated components are effective across diverse environments (Section 3.3).

## 3.1   Experimental Setup

**Benchmarks:**   We evaluate Simulus on three sample-efficiency benchmarks of different observation and action modalities: Atari 100K [28], DeepMind Control Suite (DMC) Proprioception 500K [53], and Craftax-1M [35].

Atari 100K has become the gold standard in the literature for evaluating sample-efficient deep RL agents. The benchmark comprises a subset of 26 games. Within each game, agents must learn from visual image signal under a tightly restricted budget of 100K interactions, corresponding to roughly two hours of human gameplay.

The DeepMind Control Suite (DMC) is a set of continuous control tasks involving multiple agent embodiments ranging from simple single-joint models to complex humanoids. Here, we follow the

subset of proprioception tasks used for the evaluation of DreamerV3 [20], where observations and actions are continuous vectors. At each task, the agent's interaction budget is limited to 500K steps.

Craftax is a 2D open-world survival game benchmark inspired by Minecraft, designed to evaluate RL agents' capabilities in planning, memory, and exploration. The partially-observable environment features procedurally generated worlds where agents must gather and craft resources while surviving against hostile creatures. Observations consist of a 9×11 tile egocentric map, where each tile consists of 4 symbols, and 48 state features corresponding to state information such as inventory and health. Here, we consider the sample-efficiency oriented Craftax-1M variant which only allows an interaction budget of one million steps.

**Baselines**   On Atari-100K, we compare Simulus against DreamerV3 [20] and several methods restricted to image observations: TWM [44], STORM [60], DIAMOND [3], and REM [11]. On DMC, we compare exclusively with DreamerV3, currently the only planning-free world model method with published results on the 500K proprioception benchmark. On Craftax-1M, we compare against TWM [13], a concurrent work that proposes a Transformer based world model with a focus on the Craftax benchmark, and the baselines reported in the Craftax paper: Random Network Distillation (RND) [10], PPO [48] with a recurrent neural network (PPO-RNN), and Exploration via Elliptical Episodic Bonuses (E3B) [23]. As Craftax is a recent benchmark, there are no other published results in existing world models literature. Following the standard practice in the literature, we exclude planning-based methods [21, 57], as planning is an orthogonal component that operates on any given model, typically incurring significant computational overhead.

**Metrics and Evaluation**   For Atari, we report human-normalized scores (HNS), calculated as $\frac{\text{agent\_score}-\text{random\_score}}{\text{human\_score}-\text{random\_score}}$ [39]. Following the protocol of Agarwal et al. [2] and using their toolkit, we report the mean, median, interquartile mean (IQM), and optimality gap metrics with 95% stratified bootstrap confidence intervals. For DMC and Craftax, we report the raw agent returns. We use 5 random seeds per environment. In each experiment, final performance is evaluated using 100 test episodes at the end of training and the mean score is reported.

## 3.2   Results

Simulus achieves state-of-the-art performance across all three benchmarks (Figure 1). On Atari 100k, it outperforms all baselines across key metrics (Figure 4). Notably, Simulus is the first planning-free world model to reach human-level IQM and median scores, achieving superhuman performance

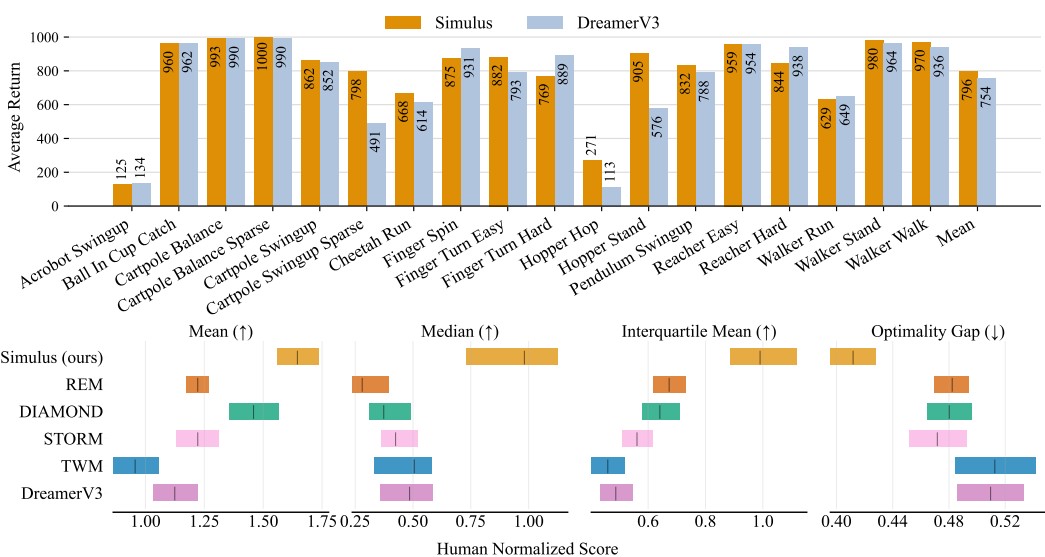

Figure 4: Results on the DeepMind Control Suite 500K Proprioception (top) and Atari 100K (bottom) benchmarks.

on **13** out of 26 games (Table 9, Appendix B). Building on REM, these gains are attributed to the integration of the proposed components, demonstrating their combined effectiveness.

**Effectiveness in continuous environments** Figure 4 provides compelling evidence that token-based architectures can perform well in continuous domains—even with compact vocabularies: Simulus consistently matches DreamerV3 across most tasks and slightly outperforms it on average.

**Effectiveness in environments with multi-modal observations** We evaluate multi-modality performance in Craftax, as it combines an image-like 2D grid map with a vector of features, involving multiple tokenizers ($\mathcal{V}$). Simulus maintains sample-efficiency in this multi-modal environment, outperforming both concurrent world model methods (Figure 1) and all model-free baselines (Figure 5), including exploration-focused algorithms. With 444 tokens per observation arranged into sequences of 147 embeddings, even short trajectories in Craftax contain thousands of tokens, demonstrating Simulus's efficient handling of long sequences. These findings indicate that the world model ($\mathcal{M}$) and controller ($\mathcal{C}$) maintain strong performance under multi-modal inputs when processed by the proposed modular tokenizer.

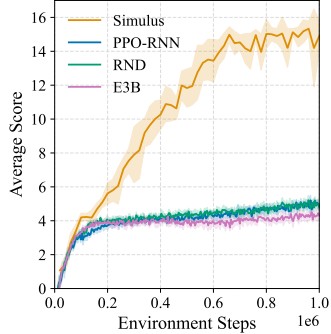

Figure 5: Craftax-1M training curves with mean and 95% confidence intervals.

### 3.3 Ablation Studies

We ablate the intrinsic rewards, prioritized replay, and regression-as-classification to demonstrate their individual contributions to Simulus's performance. In each experiment, Simulus is modified by disabling a single component. Due to limited computational resources, we consider a subset of 8 tasks for each of the Atari and DMC benchmarks, and exclude Craftax from this analysis. For Atari 100K, we used games in which significant improvements were observed. For DMC, we chose a subset that includes different embodiments. We defer the specific environment names to Appendix C.

The results are presented in Figure 6. Although all components contributed to Simulus's final performance, intrinsic rewards were especially crucial for achieving competitive performance in both benchmarks. Interestingly, the Atari 100K results indicate that combining all three components yields a significantly stronger algorithm. These findings also suggest that both prioritized world model replay and regression-as-classification enhance the effectiveness of intrinsic rewards.

More broadly, the results in Figure 6 demonstrate that encouraging the controller to explore regions of high epistemic uncertainty through intrinsic rewards significantly improves its performance in world model agents, even in reward-rich environments. This observation is non-trivial in a sample-efficient setting, where the limited interaction budget makes model-driven exploration particularly costly, as it consumes resources that could otherwise be used for task-related exploration during data collection. The latter type of exploration aims to collect new information about the true reward signal, which defines the task and its success metric. On the other hand, model-driven exploration may guide the controller towards environment regions that are irrelevant to the task at hand.

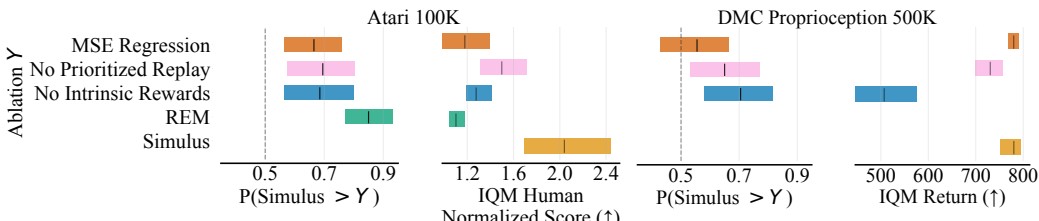

Figure 6: Ablations results on the Atari-100K and DeepMind Control Proprioception 500K benchmarks. A subset of 8 games was used for each ablation.

## 4 Related Work

**Offline Multi-Modal Methods**  Large-scale token sequence models for multi-modal agent trajectories have been proposed in [33, 34, 43, 47]. Gato [43] and TDM [47] tokenize inputs via predefined transformations, while Unified IO [33, 34] leverages pretrained models. These methods do not learn control through RL but rely on expert data. They also use massive models—with billions of parameters, large vocabularies, and significantly more data and compute than sample-efficient world models. Consequently, it remains unclear whether their design choices would be effective in online, sample-efficient settings with non-stationary and limited data.

**World Model Agents**  Model-based agents that learn policies solely from simulated data generated by a learned world model were introduced by Ha and Schmidhuber [17], followed by the influential Dreamer family [18–20]. Dreamer jointly optimizes its representation and recurrent world models via a KL divergence between learned prior and posterior distributions, leading to interfering objectives (Appendix B.1) and a complex, monolithic architecture that complicates development and scaling [59]. With the rise of Transformer architectures in language modeling [56, 9], Transformer-based Dreamer variants emerged [60, 44], alongside token-based world models (TBWMs) that treat trajectories as discrete token sequences [37, 11]. However, these methods are limited to visual environments with discrete actions (e.g., Atari 100K), leaving their performance in other modalities uncertain. Recently, DIAMOND [3], a diffusion world model inspired by advances in generative modeling [45], was introduced. While it generates visually compelling outputs, it remains limited to visual domains.

**Intrinsically Motivated World Model Agents**  Although intrinsic motivation (IM) has been extensively studied [41, 10, 22, 5, 4], its use in world models typically involves an exploration pretraining phase followed by limited task-specific fine-tuning [49, 36, 30]. While combining IM with prioritized replay has shown promise [30], it remains unexplored in standard sample-efficiency settings with external rewards.

**Large-Scale Video World Models**  Building on recent advances in video generative modeling [25, 7, 8], recent works have introduced large-scale video world models [14, 1, 12, 54], trained offline on extensive pre-collected data to predict future frames. However, these models do not address control learning, particularly RL. While recent efforts aim to bridge this gap [59], they remain confined to visual environments and lack comprehensive empirical evaluation.

## 5 Limitations and Future Work

Here, we briefly highlight several limitations of this work. First, although the feature quantization approach for tokenizing continuous vectors showed promise, it leads to excessive sequence lengths. We believe that more efficient solutions can be found for dealing with continuous inputs. Second, due to the scarcity of rich multi-modal RL benchmarks, we could not extensively explore diverse modality combinations in our experiments. Lastly, token-based world model agents remain significantly slower to train than other baselines in sample-efficient RL. Nonetheless, their modular design enables faster policy inference as the controller is independent of the world model.

## 6 Conclusions

In this paper, we demonstrated the effectiveness of several underutilized techniques for improving world model agents. A modular multi-modality tokenization framework broadens their applicability across diverse domains, while intrinsic motivation, prioritized world model replay, and regression-as-classification enhance sample efficiency, particularly when combined. These techniques were incorporated into a token-based world model, yielding Simulus. Extensive experiments show that Simulus achieves state-of-the-art performance across diverse benchmarks, including visual, continuous, and symbolic domains. It outperforms all baselines on key metrics in both Atari-100K and the challenging Craftax benchmarks. Ablations further highlight the individual contribution of each component. We hope that the highly modular design of Simulus, along with the released code and model weights, provides a strong foundation for future work.

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

# A Models and Hyperparameters

## A.1 Hyperparameters

We detail shared hyperparameters in Table 1, training hyperparameters in Table 2, world model hyperparameters in Table 3, and controller hyperparameters in Table 4. Environment hyperparameters are detailed in Table 5 (Atari-100K) and Table 6 (DMC).

For the DMC benchmark, we use a lower embedding dimension (Table 3) due to the significantly lower dimensionality of its observations compared to other benchmarks. As in prior work (e.g., DreamerV3), we adopt a smaller model for this setting. Note that the reduced number of Retention heads ensures a consistent head dimensionality (64).

Additionally, we used a limited decay range in DMC (Table 3) as observations effectively represent full MDP states, eliminating the need for long-term memory. By constraining the decay range, we explicitly encode this inductive bias into the model.

In the Craftax benchmark, we reduce the number of layers (Table 3) to lower computational cost. The interaction budget in Craftax is 1M steps, resulting in a particularly expensive training process. Here, we increase the decay range as an inductive bias to encourage long-term memory.

The weighting of intrinsic versus extrinsic rewards (Table 4) varies across benchmarks due to differences in reward structure and scale. For instance, DMC provides dense rewards with typical task scores reaching around 1000, whereas Craftax has extremely sparse rewards, with cumulative scores rarely exceeding 20—even over thousands of steps.

**Tuning**   All remaining hyperparameters were tuned empirically or based on REM [11], with minimal impact on training cost and adjusted primarily for performance. Due to limited computational resources, we were unable to conduct extensive tuning, and we believe that further optimization could improve Simulus's performance.

Table 1: Shared hyperparameters.

| Description | Symbol | Value |
|---|---|---|
| Eval sampling temperature | | 0.5 |
| Optimizer | | AdamW |
| Learning rate ($\mathcal{V}, \mathcal{M}, \mathcal{C}$) | | (1e-4, 2e-4, 2e-4) |
| AdamW $\beta_1$ | | 0.9 |
| AdamW $\beta_2$ | | 0.999 |
| Gradient clipping threshold ($\mathcal{V}, \mathcal{M}, \mathcal{C}$) | | (10, 3, 3) |
| Weight decay ($\mathcal{V}, \mathcal{M}, \mathcal{C}$) | | (0.01, 0.05, 0.01) |
| Prioritized replay fraction | $\alpha$ | 0.3 |
| Prioritized replay initial loss value | $\nu_0$ | 10 |
| $\mathcal{M}$ ensemble size | $N_{\text{ens}}$ | 4 |
| HL-Gauss num bins | | 128 |
| Label smoothing | $\sigma$ | $\frac{3}{4}$bin_width $= 0.1758$ |

Table 2: Training hyperparameters.

| Description | Symbol | Atari-100K | DMC | Craftax |
|---|---|---|---|---|
| Horizon | $H$ | 10 | 20 | 20 |
| Observation sequence length | $K$ | 64 | 3-24 | 147 |
| Action sequence length | $K_a$ | 1 | 1-6 | 1 |
| Tokenizer vocabulary size | $N$ | 512 | 125 | (37,5,40,20,4,125) |
| Epochs | | 600 | 1000 | 10000 |
| Experience collection epochs | | 500 | 1000 | 10000 |
| Environment steps per epoch | | 200 | 500 | 100 |
| Batch size ($\mathcal{V}, \mathcal{M}, \mathcal{C}$) | | (128, 32, 128) | (-, 16, 128) | (-, 8, 128) |
| Training steps per epoch ($\mathcal{V}, \mathcal{M}, \mathcal{C}$) | | (200, 200, 80) | (-, 300, 100) | (-, 100, 50) |
| Training start after epoch ($\mathcal{V}, \mathcal{M}, \mathcal{C}$) | | (5, 25, 50) | (-, 15, 20) | (-, 250, 300) |

Table 3: World model ($\mathcal{M}$) hyperparameters.

| Description | Symbol | Atari-100K | DMC | Craftax |
|---|---|---|---|---|
| Number of layers | | 10 | 10 | 5 |
| Number of heads | | 4 | 3 | 4 |
| Embedding dimension | $d$ | 256 | 192 | 256 |
| Dropout | | 0.1 | 0.1 | 0.1 |
| Retention decay range | $[\eta_{\min}, \eta_{\max}]$ | [4, 16] | [2, 2] | [8, 40] |

Table 4: Actor-critic ($\mathcal{C}$) hyperparameters.

| Description | Symbol | Atari-100K | DMC | Craftax |
|---|---|---|---|---|
| Environment reward weight | $w^{\text{ext}}$ | 1 | 1 | 100 |
| Intrinsic reward weight | $w^{\text{int}}$ | 1 | 10 | 1 |
| Encoder MLP ($g_\psi$) hidden layer sizes | | [512] | [384] | [512, 512] |
| Shared backbone | | True | False | True |
| Number of quantization values (continuous actions) | | | 51 | |
| (2D) Categoricals embedding dimension | | | | 64 |

Table 5: Atari 100K hyperparameters.

| Description | Symbol | Value |
|---|---|---|
| Frame resolution | | $64 \times 64$ |
| Frame Skip | | 4 |
| Max no-ops (train, test) | | (30, 1) |
| Max episode steps (train, test) | | (20K, 108K) |
| Terminate on live loss (train, test) | | (No, Yes) |

Table 6: DeepMind Control Suite Proprioception hyperparameters.

| Description | Symbol | Value |
|---|---|---|
| Action repeat | | 2 |

## A.2 The Representation Module $\mathcal{V}$

### A.2.1 Image Observations

Image observations are tokenized using a vector-quantized variational auto-encoder (VQ-VAE) [55, 15]. A VQ-VAE comprises a convolutional neural network (CNN) encoder, an embedding table $\mathbf{E} \in \mathbb{R}^{n \times d}$, and a CNN decoder. Here, the size of the embedding table $n$ determines the vocabulary size.

The encoder's output $\mathbf{h} \in \mathbb{R}^{W \times H \times d}$ is a grid of $W \times H$ multi-channel vectors of dimension $d$ that encode high-level learned features. Each such vector is mapped to a discrete token by finding the closest embedding in $\mathbf{E}$:

$$z = \arg \min_i \|\mathbf{h} - \mathbf{E}(i)\|,$$

where $\mathbf{E}(i)$ is the $i$-th row of $\mathbf{E}$. To reconstruct the original image, the decoder first maps $\mathbf{z}$ to their embeddings using $\mathbf{E}$. During training, the straight-through estimator [6] is used for backpropagating the learning signal from the decoder to the encoder: $\hat{\mathbf{h}} = \mathbf{h} + \mathrm{sg}(\mathbf{E}_z - \mathbf{h})$. The architecture of the encoder and decoder models is presented in Table 7.

The optimization objective is given by

$$\mathcal{L}(\mathrm{enc}, \mathrm{dec}, \mathbf{E}) =$$
$$\|\mathbf{o} - \mathrm{dec}(z)\|_2^2 + \|\mathrm{sg}(\mathrm{enc}(\mathbf{o})) - \mathbf{E}(z)\|_2^2 + \|\mathrm{sg}(\mathbf{E}(z) - \mathrm{enc}(\mathbf{o})\|_2^2 + \mathcal{L}_{\mathrm{perceptual}}(\mathbf{o}, \mathrm{dec}(z)),$$

where $\mathcal{L}_{\mathrm{perceptual}}$ is a perceptual loss [27, 32], proposed in [37].

Crucially, the learned embedding table $\mathbf{E}$ is used for embedding the (image) tokens across all stages of the algorithm.

### A.2.2 Continuous Vectors

The quantization of each feature uses 125 values (vocabulary size) in the range $[-6, 6]$, where 63 values are uniformly distributed in $[-\ln(1 + \pi), \ln(1 + \pi)]$ and the rest are uniformly distributed in the remaining intervals.

Table 7: The encoder and decoder architectures of the VQ-VAE model. "Conv(a,b,c)" represents a convolutional layer with kernel size $a \times a$, stride of $b$ and padding $c$. A value of $c =$ Asym. represents an asymmetric padding where a padding of 1 is added only to the right and bottom ends of the image tensor. "GN" represents a GroupNorm operator with 8 groups, $\epsilon = 1e - 6$ and learnable per-channel affine parameters. SiLU is the Sigmoid Linear Unit activation [24, 42]. "Interpolate" uses PyTorch's interpolate method with scale factor of 2 and the "nearest-exact" mode.

| Module | Output Shape |
|---|---|
| Encoder | |
| Input | $3 \times 64 \times 64$ |
| Conv(3, 1, 1) | $64 \times 64 \times 64$ |
| EncoderBlock1 | $128 \times 32 \times 32$ |
| EncoderBlock2 | $256 \times 16 \times 16$ |
| EncoderBlock3 | $512 \times 8 \times 8$ |
| GN | $512 \times 8 \times 8$ |
| SiLU | $512 \times 8 \times 8$ |
| Conv(3, 1, 1) | $256 \times 8 \times 8$ |
| EncoderBlock | |
| Input | $c \times h \times w$ |
| GN | $c \times h \times w$ |
| SiLU | $c \times h \times w$ |
| Conv(3, 2, Asym.) | $2c \times \frac{h}{2} \times \frac{w}{2}$ |
| Decoder | |
| Input | $256 \times 8 \times 8$ |
| BatchNorm | $256 \times 8 \times 8$ |
| Conv(3, 1, 1) | $256 \times 8 \times 8$ |
| DecoderBlock1 | $128 \times 16 \times 16$ |
| DecoderBlock2 | $64 \times 32 \times 32$ |
| DecoderBlock3 | $64 \times 64 \times 64$ |
| GN | $64 \times 64 \times 64$ |
| SiLU | $64 \times 64 \times 64$ |
| Conv(3, 1, 1) | $3 \times 64 \times 64$ |
| DecoderBlock | |
| Input | $c \times h \times w$ |
| GN | $c \times h \times w$ |
| SiLU | $c \times h \times w$ |
| Interpolate | $c \times 2h \times 2w$ |
| Conv(3, 1, 1) | $\frac{c}{2} \times 2h \times 2w$ |

### A.3 The World Model $\mathcal{M}$

**Embedding Details**  Each token in $\mathbf{z}^{(i)}$ of each modality is mapped to a $d$-dimensional embedding vector $\mathbf{X}^{(i)}$ using the embedding (look-up) table $\mathbf{E}^{(i)}$ of modality $\kappa_i$. The embedding vector that corresponds to token $z$ is simply the $z$-th row in the embedding table. Formally, $\mathbf{x}_{t,j}^{(i)} = \mathbf{E}^{(i)}(l)$, $l = z_{t,j}^{(i)}$ where $\mathbf{E}(l)$ refers to the $l$-th row in $\mathbf{E}$. In the special case of 2D categorical inputs, $\mathbf{x}_{t,j}^{(i)} = \frac{1}{C} \sum_{n=1}^{C} \mathbf{E}_n^{(i)}(l_n)$, $l_n = z_{t,j,n}^{(i)}$ where $C$ is the number of channels and $i$ is the index of the 2D categorical modality in $\kappa$.

To concatenate the embeddings, we use the following order among the modalities: images, continuous vectors, categorical variables, and 2D categoricals.

**Prediction Heads**  Each prediction head in $\mathcal{M}$ is a multi-layer perceptron (MLP) with a single hidden layer of dimension $2d$ where $d$ is the embedding dimension.

**Epistemic Uncertainty Estimation**  Working with discrete distributions enables efficient entropy computation and ensures that the ensemble disagreement term $\delta_t$ is bounded by $\frac{1}{|\mathbf{z}_t|} \sum_{z \in \mathbf{z}_t} \log(\text{vocab\_size}(z))$.

#### A.3.1 Optimization

For each training example in the form of a trajectory segment in token representation $\tau = \mathbf{z}_1, \mathbf{z}_1^{\mathrm{a}}, \ldots, \mathbf{z}_H, \mathbf{z}_H^{\mathrm{a}}$, the optimization objective is given by

$$\mathcal{L}_{\mathcal{M}}(\theta, \phi, \tau) = \sum_{t=1}^{H} \mathcal{L}_{\text{obs}}(\theta, \mathbf{z}_t, p_\theta(\hat{\mathbf{z}}_t | \mathbf{Y}_t^{\mathrm{u}})) + \mathcal{L}_{\text{reward}}(\theta, r_t, \hat{r}_t) - \log(p_\theta(d_t | \mathbf{Y}_t^{\mathrm{u}}))$$
$$+ \sum_{i=1}^{N_{\text{ens}}} \mathcal{L}_{\text{obs}}(\phi_i, \mathbf{z}_t, p_{\phi_i}(\hat{\mathbf{z}}_t | \text{sg}(\mathbf{Y}_t^{\mathrm{u}}))),$$

where

$$\mathcal{L}_{\text{obs}}(\theta, \mathbf{z}_t, p_\theta(\hat{\mathbf{z}}_t | \mathbf{Y}_t^{\mathrm{u}})) = -\frac{1}{K} \sum_{i=1}^{K} \log p_\theta(z_i | \mathbf{y}_i)$$

is the average of the cross-entropy losses of the individual tokens, and $\mathcal{L}_{\text{reward}}(\theta, r_t, \hat{r}_t)$ is the $\mathcal{L}_{\text{HL-Gauss}}$ loss with the respective parameters of the reward head. Here, $\mathbf{y}_i$ is the vector of $\mathbf{Y}_t^{\mathrm{u}}$ that corresponds to $z_i$, the $i$-th token of $\mathbf{z}_t$.

#### A.3.2 Retentive Networks

Retentive Networks (RetNet) [51] are sequence model architectures with a Transformer-like structure [56]. However, RetNet replaces the self-attention mechanism with a linear-attention [29] based Retention mechanism. At a high level, given an input sequence $\mathbf{X} \in \mathbb{R}^{|\mathbf{X}| \times d}$ of $d$-dimensional vectors, the Retention operator outputs

$$\text{Retention}(\mathbf{X}) = (\mathbf{Q}\mathbf{K}^\top \odot \mathbf{D})\mathbf{V},$$

where $\mathbf{Q}, \mathbf{K}, \mathbf{V}$ are the queries, keys, and values, respectively, and $\mathbf{D}$ is a causal mask and decay matrix. Notably, the softmax operation is discarded in Retention and other linear attention methods. As a linear attention method, the computation can also be carried in a recurrent form:

$$\text{Retention}(\mathbf{x}_t, \mathbf{S}_{t-1}) = \mathbf{S}_t \mathbf{q}_t,$$
$$\mathbf{S}_t = \eta \mathbf{S}_{t-1} + \mathbf{v}_t \mathbf{k}_t^\top \in \mathbb{R}^{d \times d},$$

where $\eta$ is a decay factor, $\mathbf{S}_t$ is a recurrent state, and $\mathbf{S}_0 = 0$. In addition, a hybrid form of recurrent and parallel forward computation known as the chunkwise mode allows to balance the quadratic cost of the parallel form and the sequential cost of the recurrent form by processing the input as a sequence of chunks. We refer the reader to [51] for the full details about this architecture.

In our implementation, since inputs are complete observation-action block sequences $\mathbf{X}_1, \ldots, \mathbf{X}_t$, we configure the decay factors of the multi-scale retention operator in block units:

$$\eta = 1 - 2^{-\text{linspace}(\log_2(K\eta_{\min}), \log_2(K\eta_{\max}), N_h)},$$

where $\text{linspace}(a, b, c)$ is a sequence of $c$ values evenly distributed between $a$ and $b$, $N_h$ is the number of retention heads, and $\eta_{\min}, \eta_{\max}$ are hyperparameters that control the memory decay in observation-action block units.

### A.3.3 Parallel Observation Prediction (POP)

POP [11] is a mechanism for parallel generation of non-causal subsequences such as observations in token representation. It's purpose is to improve generation efficiency by alleviating the sequential bottleneck caused by generating observations a single token at a time (as done in language models). However, to achieve this goal, POP also includes a mechanism for maintaining training efficiency. Specifically, POP extends the chunkwise forward mode of RetNet to maintain efficient training of the sequence model.

To generate multiple tokens into the future at once, POP introduces a set of prediction tokens $\mathbf{u} = u_1, \ldots, u_K$ and embeddings $\mathbf{X}^{\text{u}} \in \mathbb{R}^{K \times d}$ where $K$ is the number of tokens in an observation. Each token in $\mathbf{u}$ corresponds to an observation token in $\mathbf{z}$. These tokens, and their respective learned embeddings, serve as a learned prior.

Let $\mathbf{X}_1, \ldots, \mathbf{X}_T$ be a sequence of $T$ observation-action (embeddings) blocks. Given $\mathbf{S}_{t-1}$ summarizing all key-value outer products of elements of $\mathbf{X}_{\leq t-1}$, the outputs $\mathbf{Y}^{\text{u}}$ from which the next observation tokens are predicted are given by:

$$(\cdot, \mathbf{Y}_t^{\text{u}}) = f_\theta(\mathbf{S}_{t-1}, \mathbf{X}^{\text{u}}).$$

Importantly, the recurrent state is never updated based on the prediction tokens $\mathbf{u}$ (or their embeddings). The next observation tokens $\hat{\mathbf{z}}_t$ are sampled from $p_\theta(\hat{\mathbf{z}}_t | \mathbf{Y}_t^{\text{u}})$. Then, the next action is generated by the controller, and the next observation-action block $\mathbf{X}_t$ can be processed to predict the next observation $\hat{\mathbf{z}}_{t+1}$.

To maintain efficient training, a two step computation is carried at each RetNet layer. First, all recurrent states $\mathbf{S}_t$ for all $1 \leq t \leq T$ are calculated in parallel. Although there is an auto-regressive relationship between time steps, the linear structure of $\mathbf{S}$ allows to calculate the compute-intensive part of each state in parallel and incorporate past information efficiently afterwards. In the second step, all outputs $\mathbf{Y}_t^{\text{u}}$ for all $1 \leq t \leq T$ are computed in parallel, using the appropriate states $\mathbf{S}_{t-1}$ and $\mathbf{X}^{\text{u}}$ in batch computation. Note that this computation involves delicate positional information handling. We refer the reader to [11] for full details of this computation.

## A.4 The Controller $\mathcal{C}$

**Critic** The value prediction uses 128 bins in the range $\mathbf{b} = (-15, \ldots, 15)$.

**Continuous Action Spaces** The policy network outputs $m = 51$ logits corresponding to $m$ quantization values uniformly distributed in $[-1, 1]$ for each individual action in the action vector.

### A.4.1 Input Encoding

The controller $\mathcal{C}$ operates in the latent token space. Token trajectories $\tau = \mathbf{z}_1, \mathbf{z}_1^{\mathrm{a}}, \ldots, \mathbf{z}_H, \mathbf{z}_H^{\mathrm{a}}$ are processed sequentially by the LSTM model. At each time step $t$, the network gets $\mathbf{z}_t$ as input, outputs $\pi(\mathbf{a}_t|\tau_{\leq t-1}, \mathbf{z}_t)$ and $\hat{V}^\pi(\mathbf{a}_t|\tau_{\leq t-1}, \mathbf{z}_t)$, samples an action $\mathbf{a}_t$ and then process the sampled action as another sequence element.

The processing of actions involve embedding the action into a latent vector which is then provided as input to the LSTM. Embedding of continuous action tokens is performed by first reconstructing the continuous action vector and then computing the embedding using a linear projection. Discrete tokens are embedded using a dedicated embedding table.

To embed observation tokens $\mathbf{z}$, the tokens of each modality are processed by a modality-specific encoder. The outputs of the encoders are concatenated and further processed by a MLP $g_\psi$ that combines the information into a single vector latent representation.

The image encoder is a convolutional neural network (CNN). Its architecture is given in Table 8.

Categorical variables are embedded using a learned embedding table. For 2D categoricals, shared per-channel embedding tables map tokens to embedding vectors, which are averaged to obtain a single embedding for each multi-channel token vector. For both types of categorical inputs we use 64 dimensional embeddings. The embeddings are concatenated and processed by $g_\psi$.

Table 8: The image observation encoder architecture of the actor-critic controller $\mathcal{C}$.

| Module | Output Shape |
|---|---|
| Input | $256 \times 8 \times 8$ |
| Conv(3, 1, 1) | $128 \times 8 \times 8$ |
| SiLU | $128 \times 8 \times 8$ |
| Conv(3, 1, 1) | $64 \times 8 \times 8$ |
| SiLU | $64 \times 8 \times 8$ |
| Flatten | 4096 |
| Linear | 512 |
| SiLU | 512 |

### A.4.2 Optimization

$\lambda$-returns are computed for each generated trajectory segment $\hat{\tau} = (\mathbf{z}_1, \mathbf{a}_1, \bar{r}_1, d_1, \hat{\mathbf{z}}_2, \mathbf{a}_2, \bar{r}_2, d_2, \ldots, \hat{\mathbf{z}}_H, \mathbf{a}_H, \bar{r}_H, d_H)$:

$$G_t = \begin{cases} \bar{r}_t + \gamma(1 - d_t)((1 - \lambda)\hat{V}_{t+1}^\pi + \lambda G_{t+1}) & t < H \\ \hat{V}_H^\pi & t = H \end{cases},$$

where $\hat{V}_t^\pi = \hat{V}^\pi(\hat{\tau}_{\leq t})$. These $\lambda$-returns are used as targets for critic learning. For policy learning, a REINFORCE [52] objective is used, with a normalized $\hat{V}^\pi$ baseline for variance reduction:

$$\mathcal{L}_\pi(\psi) = \mathbb{E}_\pi \left[ \sum_{t=1}^H \mathrm{sg}\left( \frac{G_t - \hat{V}_t^\pi}{\max(1, c)} \right) \log \pi(\mathbf{a}_t|\hat{\tau}_{\leq t-1}, \hat{\mathbf{z}}_t) + w_{\mathrm{ent}}\mathcal{H}(\pi(\mathbf{a}_t|\hat{\tau}_{\leq t-1}, \hat{\mathbf{z}}_t)) \right],$$

where $c$ is an estimate of the effective return scale similar to DreamerV3 [20] and $w_{\mathrm{ent}}$ is a hyper-parameter that controls the entropy regularization weight. $c$ is calculated as the difference between the running average estimators of the $97.5$ and $2.5$ return percentiles, based on a window of return estimates obtained in the last $500$ batches (imagination).

# B   Additional Results

The average per-game scores for Atari-100K are presented in Table 9. The performance profile plot
for Atari 100K is presented in Figure 7.

Table 9: Mean returns on the 26 games of the Atari 100k benchmark followed by averaged human-normalized performance metrics. Each game score is computed as the average of 5 runs with different seeds. Bold face mark the best score.

| Game | Random | Human | DreamerV3 | TWM | STORM | DIAMOND | REM | SIMULUS (ours) |
|---|---|---|---|---|---|---|---|---|
| Alien | 227.8 | 7127.7 | 959.4 | 674.6 | **983.6** | 744.1 | 607.2 | 687.2 |
| Amidar | 5.8 | 1719.5 | 139.1 | 121.8 | 204.8 | **225.8** | 95.3 | 102.4 |
| Assault | 222.4 | 742.0 | 705.6 | 682.6 | 801.0 | 1526.4 | 1764.2 | **1822.8** |
| Asterix | 210.0 | 8503.3 | 932.5 | 1116.6 | 1028.0 | **3698.5** | 1637.5 | 1369.1 |
| BankHeist | 14.2 | 753.1 | **648.7** | 466.7 | 641.2 | 19.7 | 19.2 | 347.1 |
| BattleZone | 2360.0 | 37187.5 | 12250.0 | 5068.0 | **13540.0** | 4702.0 | 11826.0 | 13262.0 |
| Boxing | 0.1 | 12.1 | 78.0 | 77.5 | 79.7 | 86.9 | 87.5 | **93.5** |
| Breakout | 1.7 | 30.5 | 31.1 | 20.0 | 15.9 | 132.5 | 90.7 | **148.9** |
| ChopperCommand | 811.0 | 7387.8 | 410.0 | 1697.4 | 1888.0 | 1369.8 | 2561.2 | **3611.6** |
| CrazyClimber | 10780.5 | 35829.4 | 97190.0 | 71820.4 | 66776.0 | **99167.8** | 76547.6 | 93433.2 |
| DemonAttack | 152.1 | 1971.0 | 303.3 | 350.2 | 164.6 | 288.1 | **5738.6** | 4787.6 |
| Freeway | 0.0 | 29.6 | 0.0 | 24.3 | 0.0 | **33.3** | 32.3 | 31.9 |
| Frostbite | 65.2 | 4334.7 | 909.4 | **1475.6** | 1316.0 | 274.1 | 240.5 | 258.4 |
| Gopher | 257.6 | 2412.5 | 3730.0 | 1674.8 | **8239.6** | 5897.9 | 5452.4 | 4363.2 |
| Hero | 1027.0 | 30826.4 | **11160.5** | 7254.0 | 11044.3 | 5621.8 | 6484.8 | 7466.8 |
| Jamesbond | 29.0 | 302.8 | 444.6 | 362.4 | 509.0 | 427.4 | 391.2 | **678.0** |
| Kangaroo | 52.0 | 3035.0 | 4098.3 | 1240.0 | 4208.0 | 5382.2 | 467.6 | **6656.0** |
| Krull | 1598.0 | 2665.5 | 7781.5 | 6349.2 | 8412.6 | **8610.1** | 4017.7 | 6677.3 |
| KungFuMaster | 258.5 | 22736.3 | 21420.0 | 24554.6 | 26182.0 | 18713.6 | 25172.2 | **31705.4** |
| MsPacman | 307.3 | 6951.6 | 1326.9 | 1588.4 | **2673.5** | 1958.2 | 962.5 | 1282.7 |
| Pong | -20.7 | 14.6 | 18.4 | 18.8 | 11.3 | **20.4** | 18.0 | 19.9 |
| PrivateEye | 24.9 | 69571.3 | 881.6 | 86.6 | **7781.0** | 114.3 | 99.6 | 100.0 |
| Qbert | 163.9 | 13455.0 | 3405.1 | 3330.8 | **4522.5** | 4499.3 | 743.0 | 2425.6 |
| RoadRunner | 11.5 | 7845.0 | 15565.0 | 9109.0 | 17564.0 | 20673.2 | 14060.2 | **24471.8** |
| Seaquest | 68.4 | 42054.7 | 618.0 | 774.4 | 525.2 | 551.2 | 1036.7 | **1800.4** |
| UpNDown | 533.4 | 11693.2 | 7567.1 | **15981.7** | 7985.0 | 3856.3 | 3757.6 | 10416.5 |
| #Superhuman (↑) | 0 | N/A | 9 | 8 | 9 | 11 | 12 | **13** |
| Mean (↑) | 0.000 | 1.000 | 1.124 | 0.956 | 1.222 | 1.459 | 1.222 | **1.645** |
| Median (↑) | 0.000 | 1.000 | 0.485 | 0.505 | 0.425 | 0.373 | 0.280 | **0.982** |
| IQM (↑) | 0.000 | 1.000 | 0.487 | 0.459 | 0.561 | 0.641 | 0.673 | **0.990** |
| Optimality Gap (↓) | 1.000 | 0.000 | 0.510 | 0.513 | 0.472 | 0.480 | 0.482 | **0.412** |

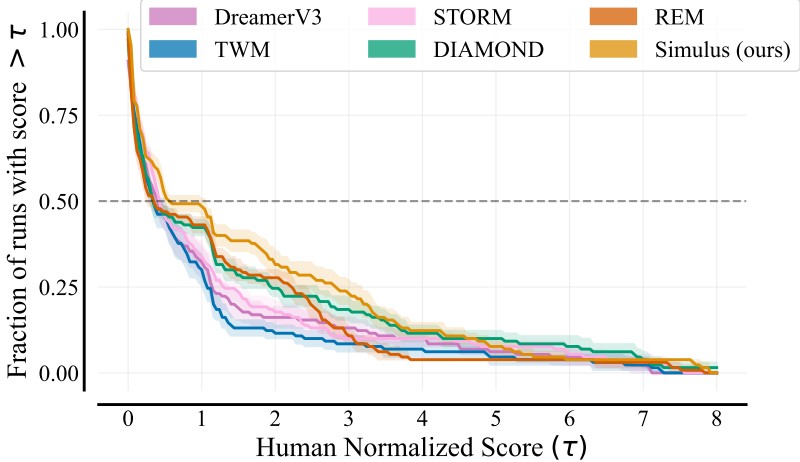

Figure 7: Performance profile. For each human-normalized score value on the x-axis, the curve of each algorithm represents the fraction of runs achieving a score greater than that value. Shaded regions denote pointwise 95% confidence intervals, computed using stratified bootstrap sampling [2].

## B.1 Interfering Objectives in RSSM Optimization

Here, we study the interplay between the objectives of a Dreamer-like world model, PWM [58], which uses a slightly modified version of the recurrent state space model (RSSM) of Dreamer. Concretely, we aim to understand whether the representation and sequence modeling objectives interfere by decoupling the optimization of the encoder-decoder models from that of the world model. We opted for this implementation due to its simplicity, fast runtime, and accessibility, as it is written in PyTorch [40].

Formally, the model consists of the following components:

$$
\begin{aligned}
\text{Encoder:} \quad & z_t \sim q_\theta(z_t | o_t), \\
\text{Decoder:} \quad & \hat{o}_t \sim p_\theta(\hat{o}_t | z_t), \\
\text{Sequence model:} \quad & h_t, x_t = f_\theta(x_{t-1}, z_{t-1}, a_{t-1}), \\
\text{Dynamics predictor:} \quad & \hat{z}_t \sim p_\theta(\hat{z}_t | h_t).
\end{aligned}
$$

We omit the reward and termination predictors and objectives for brevity. Note that in contrast to the RSSM in Dreamer, the encoder and decoder models do not depend on the recurrent state $h_t, x_t$. The optimization objective of PWM is given by

$$
\mathcal{L}(\theta) = \mathop{\mathbb{E}}_{q_\theta}\Big[\sum_{t=1}^{T} \beta_{\text{pred}}\mathcal{L}_{\text{pred}}(\theta) + \beta_{\text{dyn}}\mathcal{L}_{\text{dyn}}(\theta) + \beta_{\text{rep}}\mathcal{L}_{\text{rep}}(\theta)\Big],
$$

where $\beta_{\text{pred}}, \beta_{\text{dyn}}$, and $\beta_{\text{rep}}$ are coefficients and

$$
\begin{aligned}
\mathcal{L}_{\text{pred}}(\theta) =& \|\hat{o}_t - o_t\|_2^2, \\
\mathcal{L}_{\text{dyn}}(\theta) =& \max(1, \text{KL}[\text{sg}(q_\theta(z_t | o_t)) \| p_\theta(\hat{z}_t | h_t)]), \\
\mathcal{L}_{\text{rep}}(\theta) =& \max(1, \text{KL}[q_\theta(z_t | o_t) \| \text{sg}((p_\theta(\hat{z}_t | h_t)))]).
\end{aligned}
$$

To decouple the optimization, we modify the sequence model by introducing a stop-gradient operator on the encoder's output during world model training:

$$
h_t, x_t = f_\theta(x_{t-1}, \text{sg}(z_{t-1}), a_{t-1}).
$$

Moreover, this modification allows to train the encoder-decoder models using large batches of single frames, rather than small highly-correlated batches of long trajectories. This further highlights the flexibility and advantage of a modular design.

We compare the original PWM algorithm to its decoupled variant, PWM-decoupled, across four Atari environments: `Breakout`, `DemonAttack`, `Hero`, and `RoadRunner`. These are games where PWM performed either particularly well (`Hero` and `RoadRunner`) or poorly (`Breakout` and `DemonAttack`). Each variant is trained online from scratch on each game. The results are presented in Figure 8. In addition, we present the achieved episodic returns in Figure 9, and the reconstruction quality of example episodes in Figure 10 (PWM) and Figure 11 (PWM-decoupled).

Although our results are based on a single random seed and are limited to only four environments, we observe a consistent trend. First, the reconstruction losses are consistently and significantly lower when decoupling the optimization, while the dynamics losses are significantly higher. This suggests that the objectives are interfering.

Second, we observe similar or better episodic returns (Figure 9) using the decoupled optimization, suggesting that the higher dynamics loss might not lead to worse world modeling performance in practice. Note that a higher dynamics loss in this case does not necessarily mean worse performance, as for example multiple discrete combinations could represent the same or similar frame. Thus, when the dynamics model fails to predict a specific combination, it leads to high loss values while the underlying representations are accurate.

Lastly, we report that similar trends were observed when training only the world model in an offline, supervised-learning fashion on pre-collected datasets. We explored this setting to eliminate complexities that may arise due to the online collection of the data.

While the presented preliminary results are noisy and limited, we believe that they uncover an interesting observation on the design and optimization of current world models.

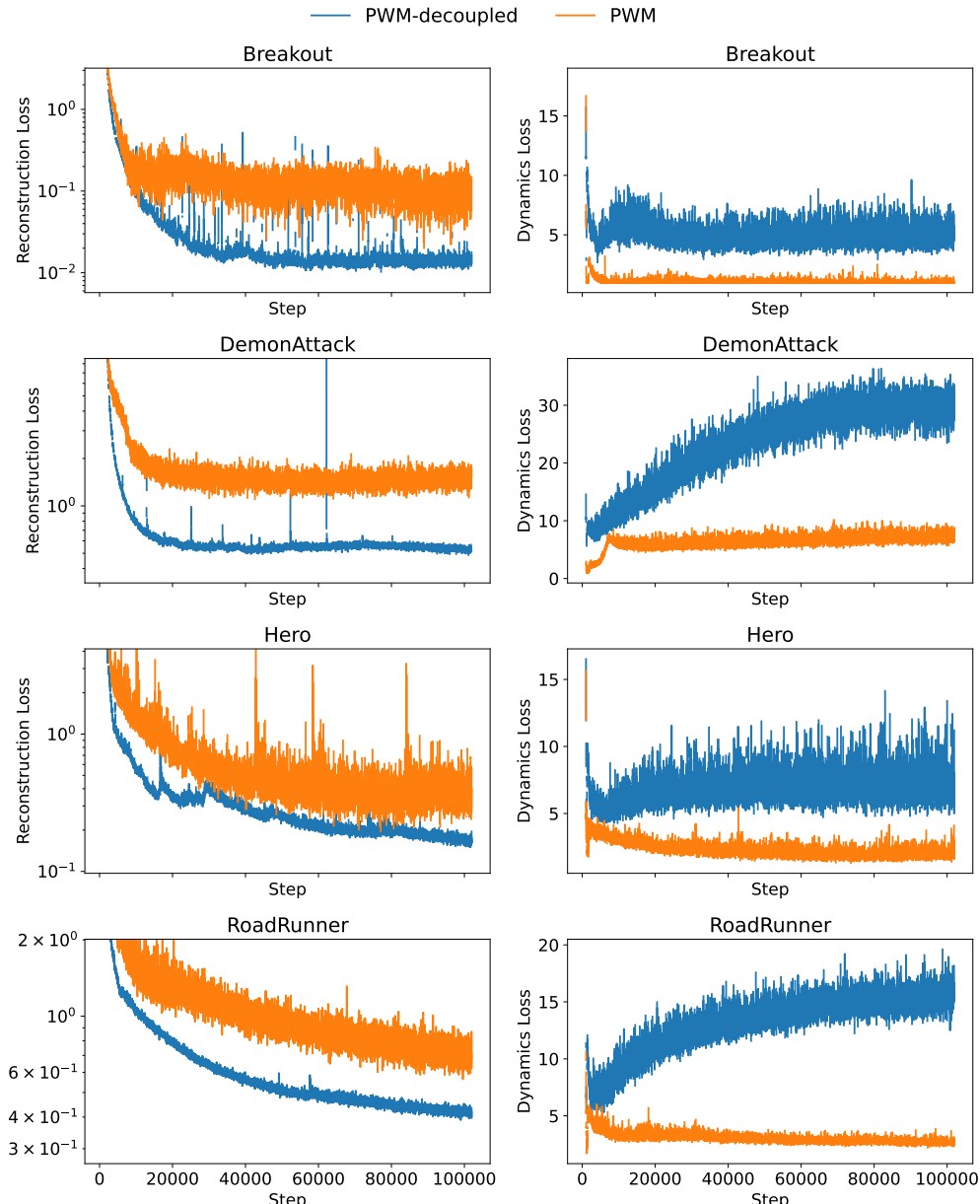

Figure 8: Reconstruction ($\mathcal{L}_{\text{pred}}$) and dynamics ($\mathcal{L}_{\text{dyn}}$) losses of PWM and PWM-decoupled on four Atari games (single seed). The first column uses a log-scaled y-axis. Decoupling the optimization objectives consistently reduces reconstruction loss while increasing dynamics loss, suggesting interference between the two objectives.

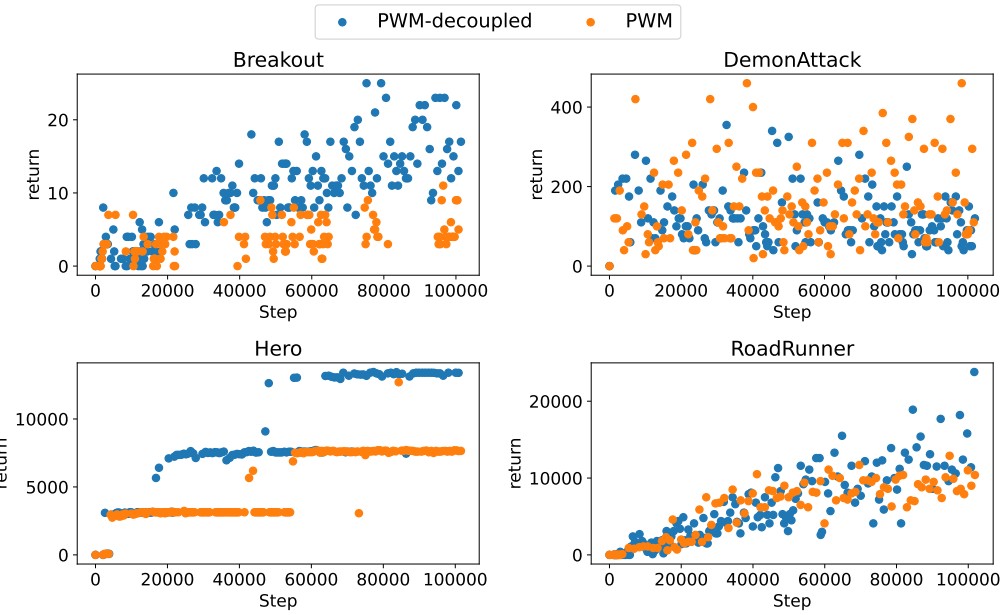

Figure 9: Agent episodic returns throughout training of PWM and PWM-decoupled on four Atari games (single seed). Each marker corresponds to a single episode.

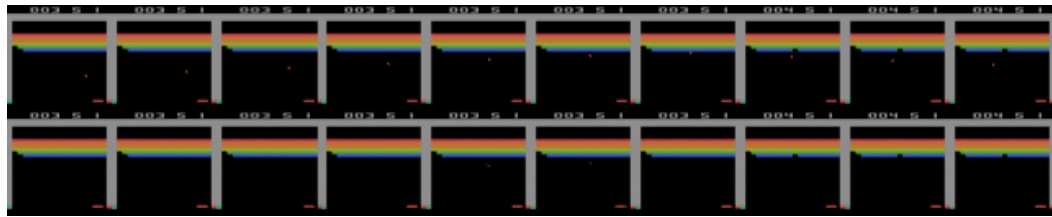

Figure 10: Ground truth (top) and reconstructed (bottom) frames from a training episode of PWM after 50K steps (half way though training). Notably, the ball is missing in most frames, suggesting the reason for its poor performance in this game.

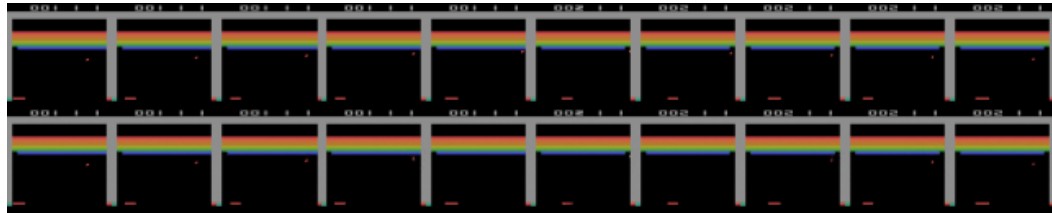

Figure 11: Ground truth (top) and reconstructed (bottom) frames from a training episode of PWM-decoupled after 50K steps (half way though training). Here, the ball is reconstructed in most frames, demonstrating the significantly improved representation performance.

## C  Implementation Details

**Ablation Studies**  For the Atari 100K benchmark, we conducted ablations on the following games: `Assault, Breakout, ChopperCommand, CrazyClimber, JamesBond, Kangaroo, Seaquest,` and `UpNDown`. For the DeepMind Control Suite, we used the tasks: `acrobot-swingup, cartpole-swingup-sparse, cheetah-run, finger-turn-hard, hopper-stand, pendulum-swingup, reacher-hard,` and `walker-run`.

**Code**  We open-source our code and trained model weights. Our code is written in Pytorch [40].

**Hardware**  All Atari and DMC experiments were performed on V100 GPUs, while for Craftax a single RTX 4090 was used.

**Run Times**  Experiments on Atari require approximately 12 hours on an RTX 4090 GPU and around 29 hours on a V100 GPU. For DMC, the runtime is about 40 hours on a V100 GPU. Craftax runs take roughly 94 hours, equivalent to 3.9 days.

**Craftax**  The official environment provides the categorical variables in one-hot encoding format. Our implementation translates these variables to integer values which can be interpreted as tokens.

**Setup in Atari Freeway**  For the Freeway environment, we adopted a modified sampling strategy where a temperature of 0.01 is used instead of the standard value of 1, following [37, 11]. This adjustment helps directing the agent toward rewarding paths. Note that alternative approaches in the literature tackle the exploration challenge through different mechanisms, including epsilon-greedy exploration schedules and deterministic action selection via argmax policies [37].

