# OpenReview forum: "Uncovering Untapped Potential in Sample-Efficient World Model Agents"
_NeurIPS.cc/2025/Conference — Submitted to NeurIPS 2025_

### Official Review · Reviewer_DVfk · 2025-06-20

**Clarity:** 3
**Significance:** 2
**Originality:** 2
**Rating:** 4
**Confidence:** 4

**Summary:**

This paper introduces Simulus, a sample-efficient, token-based world model agent. Simulus builds upon the REM agent and integrates four additional main components: (1) a modular tokenization framework to handle multi-modal observations (e.g., images, continuous vectors, symbolic grids), (2) intrinsic motivation based on epistemic uncertainty to guide exploration, (3) a prioritized replay mechanism for world model training, and (4) a regression-as-classification approach for more robust reward and value prediction. The authors conduct an empirical evaluation across three diverse and challenging benchmarks: Atari 100K, DeepMind Control Suite 500K, and Craftax-1M. The results demonstrate that Simulus achieves state-of-the-art performance across all benchmarks. Furthermore, ablation studies are presented to validate the individual contribution and synergistic effects of the integrated components.

**Questions:**

* The authors decided to discretize the continuous action space for the DMC tasks. Could you provide more insight into the potential limitations of this approach, especially for more complex robotics tasks that might require finer motor control than the ones tested?
* The authors write in the Appendix that the DMC experiments for Simulus take around 40h. Is this the run time for a single task? Compared to DreamerV3 which has a similar performance but takes only around 3h this is much longer.

**Ethical Concerns:**

["NO or VERY MINOR ethics concerns only"]

**Final Justification:**

While limited computational resources should not prevent researchers from conducting the additional experiments necessary to substantiate their claims—particularly researchers could perform more experiments needed to show the point of their method and target a later venue instead of rushing to an earlier venue—I believe the authors have nonetheless provided sufficient empirical evidence. Specifically, their integration of intrinsic motivation, prioritized experience replay, and RaC yields measurable performance improvements across a reasonable range of tasks.

**Limitations:**

The authors mention that the scarcity of multi-modal environments prevented them to evaluate further experiments with multi-modal environments. Many robotic tasks provide multi-modal observations; thus I do not fully agree with them.

**Paper Formatting Concerns:**

None so far.

**Quality:**

3

**Strengths And Weaknesses:**

### Strengths
* The paper is well-written, organized, and easy to follow. The methodology is explained in sufficient detail, and the motivations behind the design choices are well-articulated. A significant strength of this work is its adherence to modern standards for empirical evaluation in reinforcement learning. The authors report not just mean scores but also Interquartile Mean (IQM), median, and optimality gap metrics, using stratified bootstrap confidence intervals as recommended by Agarwal et al.. This provides a much more nuanced and reliable picture of performance compared to simple point estimates.
* The effectiveness of Simulus is demonstrated on three distinct and relevant benchmarks, each highlighting a different capability: visual control (Atari), continuous control from proprioception (DMC), and multi-modal reasoning (Craftax). The agent is compared against a comprehensive set of strong and recent baselines, including DreamerV3, STORM, etc., which makes the claimed state-of-the-art results highly convincing.
* The paper successfully integrates several known, powerful techniques (intrinsic motivation, prioritized replay, RaC) into a unified agent. The ablation studies compellingly demonstrate that these components not only contribute individually but also act synergistically to achieve the final reported performance.

### Weaknesses
* The primary weakness of the paper is its limited originality from a methodological standpoint. The proposed agent, Simulus, is framed as an extension of REM, and the core improvements stem from integrating existing, well-established techniques. Intrinsic motivation for exploration, prioritized replay, and RaC for value learning are all powerful but not novel ideas. While the successful engineering and integration are commendable, the work does not introduce a new fundamental algorithm or mechanism. Since the method does methodically nothing novel, a slightly stronger experimental section would strengthen the paper more to be a good paper (e.g. for the DMC tasks, which feature continuous action spaces, the authors employ a discretization strategy. I would be curious how the method performs in harder tasks with a continuous action spaces).
* A key contribution is the modular multi-modality tokenization framework. However, of the three benchmarks used, only Craftax truly requires multi-modal input processing. To make the claim of general multi-modality more compelling, the paper would benefit from evaluation on additional multi-modal environments. The authors themselves note the scarcity of such benchmarks as a limitation with which I do not fully agree with. E.g. a simple robotic setup can be multi-modal where the agent gets the proprioceptive sensor data and RGB images as input (e.g. see [ManiSkill](https://github.com/haosulab/ManiSkill)).

---

> ### Author Rebuttal · Authors · 2025-07-30
>
> We thank reviewer DVfk for their time and effort in reviewing our paper and for their constructive and valuable feedback.
>
> We appreciate that reviewer DVfk acknowledged the clarity of our presentation, rigorous evaluation practices, strong baseline comparisons, and the effective integration of known techniques, supported by insightful ablations.
>
> #### Weaknesses:
>
> `W1`:
> > The primary weakness of the paper is its limited originality from a methodological standpoint. ... the work does not introduce a new fundamental algorithm or mechanism. Since the method does methodically nothing novel, a slightly stronger experimental section would strengthen the paper ...
>
> `R1`: **Regarding the limited originality concern**, while new fundamental algorithms or mechanisms are typical indicators of novelty, demonstrating novel insights using combinations of existing methods (e.g., that combining intrinsic motivation, prioritized world-model replay, and RaC significantly boosts the performance of world model agents in sample-efficiency settings, even in dense-reward environments) can be equally valuable and impactful, as such combinations often require significant effort and insight.
> A notable example is the influential Rainbow algorithm [1] (3238 citations), which integrates known techniques to achieve strong performance. Rainbow gained significant influence despite combining existing techniques and lacking algorithmic novelty.
> We believe that our work belongs to the category of "Rainbow-style" empirical contributions, and that the absence of new algorithmic mechanisms should not be considered a weakness, as the paper offers important empirical insights instead.
>
>
> **Regarding the stronger empirical section concern**, similar published works in the world model agents literature (RL) evaluate their methods on 1-2 benchmarks, typically only Atari 100K or only continuous control [2][3][4][5][6][7]. Despite our compute limitations and disadvantage (see below), we managed to include 3 benchmarks. Hence, our empirical section is evidently stronger than the standard in the literature and we believe that it should not be considered as one of our work's weaknesses, but rather a strength.
>
> In addition, when choosing benchmarks, several factors affected our decisions.
> First, the available baselines and compute. As a small and resource-limited lab, we conducted most of our research on a single RTX 4090 GPU. Full benchmark evaluations were performed only twice, toward the end of the project, on V100 GPUs. As a result, a major factor in our benchmark selection process was the availability of external baseline results, as we could not afford to evaluate other algorithms ourselves.
> In addition, we made our best effort to include as many benchmarks as we could afford within our budget, while also maintaining diverse benchmark modalities.
> As DreamerV3 is arguably the most significant baseline with existing results on DMC, and since DMC is already an established and popular benchmark, we decided to use DMC for continuous control tasks.
>
> While exploring harder continuous control tasks can be valuable, we believe that it will not significantly strengthen the main contribution of our work which is the insight that the underexplored combination of intrinsic motivation, prioritized WM replay, and RaC is highly effective in sample-efficient world model agents, even in environments with dense rewards.
>
>
>
> [1] Hessel, Matteo, et al. "Rainbow: Combining improvements in deep reinforcement learning." Proceedings of the AAAI conference on artificial intelligence. Vol. 32. No. 1. 2018.
>
> [2] Alonso, E., Jelley, A., Micheli, V., Kanervisto, A., Storkey, A. J., Pearce, T., & Fleuret, F. (2024). Diffusion for world modeling: Visual details matter in atari. Advances in Neural Information Processing Systems, 37, 58757-58791.
>
> [3] Zhang, W., Wang, G., Sun, J., Yuan, Y., & Huang, G. (2023). Storm: Efficient stochastic transformer based world models for reinforcement learning. Advances in Neural Information Processing Systems, 36, 27147-27166.
>
> [4] Cohen, L., Wang, K., Kang, B., & Mannor, S. Improving Token-Based World Models with Parallel Observation Prediction. In Forty-first International Conference on Machine Learning.
>
> [5] Georgiev, I., Giridhar, V., Hansen, N., & Garg, A. PWM: Policy Learning with Multi-Task World Models. In The Thirteenth International Conference on Learning Representations.
>
> [6] Micheli, V., Alonso, E., & Fleuret, F. Transformers are Sample-Efficient World Models. In The Eleventh International Conference on Learning Representations.
>
> [7] Hansen, N.A., Su, H. &amp; Wang, X.. (2022). Temporal Difference Learning for Model Predictive Control. Proceedings of the 39th International Conference on Machine Learning, in Proceedings of Machine Learning Research 162:8387-8406.
>
> `W2`:
> > A key contribution is the modular multi-modality tokenization framework. However, of the three benchmarks used, only Craftax truly requires multi-modal input processing. To make the claim of general multi-modality more compelling, the paper would benefit from evaluation on additional multi-modal environments. The authors themselves note the scarcity of such benchmarks as a limitation with which I do not fully agree with. E.g. a simple robotic setup can be multi-modal where the agent gets the proprioceptive sensor data and RGB images as input (e.g. see ManiSkill).
>
> `R2`: Regarding the second point, i.e., that a simple robotic setup can be multi-modal by including both the RGB and the proprioceptive signals, note that since each of these signals is sufficient for solving the task, an agent can essentially rely on one and ignore the other. Hence, positive results on such a benchmark are not truly indicative of multi-modality capabilities. For this reason, we believe that such a benchmark would not be adequate in our case, unfortunately. That said, we truly appreciate the creative direction and the reviewer’s intent to help us improve our multi-modality evaluation.
>
> Regarding the first point, we agree that extensive evaluation on rich multi-modal environments remains a limitation, as we also acknowledged in our limitations section. However, our notion of a "multi-modality tokenization framework" also encompasses the ability to handle diverse *uni-modal* environments. Under our three-benchmark budget, already more comprehensive than the 1–2 benchmarks typical in this literature, we covered a wide range of modalities.
> Given that standard and well-established sample-efficiency benchmarks like Atari 100K and DMC are uni-modal, and since such environments are far more prevalent, demonstrating strong performance across them is arguably more important, and also provides concrete support for the framework’s multi-modality capabilities.
>
> Therefore, while we recognize the limitation, we believe the lack of additional multi-modal tasks should not be considered a major concern.
>
> #### Questions:
>
> `Q1`:
> > The authors decided to discretize the continuous action space for the DMC tasks. Could you provide more insight into the potential limitations of this approach, especially for more complex robotics tasks that might require finer motor control than the ones tested?
>
> `A1`: Discretizing continuous actions does not inherently limit controller precision, as this can be compensated by a shorter control interval or higher control frequency. However, there is a clear trade-off: large action spaces allow lower decision frequency but increase exploration cost, while smaller action spaces require more frequent decisions, increasing computational load. In our experiments, we did not encounter any issues related to action discretization.
>
> A key insight in the context of world models is that the size of the vocabulary controls not only reconstruction quality but also the ***complexity of the dynamics modeling task***, i.e., how difficult it is for the world model to learn and predict transitions. We found that using smaller vocabularies significantly improves sample efficiency, not only in continuous control but more broadly. While this may seem intuitive, we believe it is worth explicitly including in the appendix of our next revision, especially as it was also highlighted by other reviewers. We thank reviewer DVfk for raising this point.
>
> `Q2`:
> > The authors write in the Appendix that the DMC experiments for Simulus take around 40h. Is this the run time for a single task? Compared to DreamerV3 which has a similar performance but takes only around 3h this is much longer.
>
> `A2`: Training Simulus on DMC take around 40h ***on a V100 GPU***, which is equivalent to roughly 20h on an A100 GPU, with which DreamerV3 was trained. In addition, the training of DreamerV3 on proprioception tasks was reported to last 0.3 (A100) GPU days, which is equivalent to 7.2 hours. Hence, the difference (20h vs 7.2h) is not as significant as mentioned.
>
> Furthermore, as we mentioned in our limitations section, we believe that the current approach of discretizing individual features (adopted from existing literature) is inefficient, as it leads to excessive sequence lengths. Exploring more efficient solutions requires substantial efforts and is largely orthogonal to the contribution of our paper, hence we left it for future works. That said, we expect that more efficient tokenization strategies could substantially reduce the training cost of Simulus. Importantly, we believe this current inefficiency should not detract from the significance of our empirical findings.
>
> #### Limitations:
> > The authors mention that the scarcity of multi-modal environments prevented them to evaluate further experiments with multi-modal environments. Many robotic tasks provide multi-modal observations; thus I do not fully agree with them.
>
> We believe this concern has been addressed in `R2` and kindly ask you to also consider our compute limitations and the supporting arguments outlined in the second paragraph of `R1`.

---

> ### Comment · Reviewer_DVfk · 2025-08-05
>
> > Training Simulus on DMC take around 40h on a V100 GPU, which is equivalent to roughly 20h on an A100 GPU, with which DreamerV3 was trained. In addition, the training of DreamerV3 on proprioception tasks was reported to last 0.3 (A100) GPU days, which is equivalent to 7.2 hours. Hence, the difference (20h vs 7.2h) is not as significant as mentioned.
>
> The training times reported in the DreamerV3 paper are mixed up—the durations for visual control and proprioceptive control tasks should swapped. As a result, DreamerV3 requires only approximately 2.4 hours of training on proprioceptive tasks. Hence, the difference is still significant. Nonetheless, this reviewer appreciates the additional insights provided regarding potential possibilities for further training speed improvements.
>
> > A notable example is the influential Rainbow algorithm [1] (3238 citations), which integrates known techniques to achieve strong performance. Rainbow gained significant influence despite combining existing techniques and lacking algorithmic novelty. We believe that our work belongs to the category of "Rainbow-style" empirical contributions, and that the absence of new algorithmic mechanisms should not be considered a weakness, as the paper offers important empirical insights instead.
>
> While I fully agree with the underlying point, the overall tone and presentation of the Rainbow manuscript differ from that of the authors’ submission. Nonetheless, the authors convincingly demonstrate that the integration of intrinsic motivation, prioritized experience replay, and RaC leads to performance improvements across a reasonable number of tasks. In light of this, I will adjust my score accordingly.

---

> > ### Author Response · Authors · 2025-08-05
> >
> > We thank reviewer DVfk for their thoughtful response and their recognition of our empirical contributions.

---

### Official Review · Reviewer_nS4G · 2025-07-03

**Clarity:** 3
**Significance:** 2
**Originality:** 2
**Rating:** 4
**Confidence:** 2

**Summary:**

In this paper,  the author presents Simulus, a modular Tag-Based World Model agent designed to improve sample efficiency in reinforcement learning. Based on REM, Simulus integrates four main components: multimodal tokenization, intrinsic motivation for uncertainty-driven exploration, prioritized world model regeneration, and regression classification for reward prediction. It supports multiple observation modalities, including images, continuous vectors, and symbolic grids. Simulus achieves state-of-the-art performance in three benchmarks, Atari 100K, DeepMind Control Suite, and Craftax-1M, without relying on planning. Ablation studies confirm that each component makes a meaningful contribution,n and their combination produces a powerful synergy. Although training takes longer than some benchmarks, Simulus' modular design enables efficient inference and provides a solid foundation for future research.

**Questions:**

- Can the authors quantitatively report the computational cost, such as GPU hours or FLOPs? The observed improvement may be because of the increased computation rather than the method itself, which could be unfair to the baselines. One way to ensure fairness would be to extend the baseline models' training steps to match the total computational budget.

- Regarding prioritized replay, how sensitive is the model performance to the hyperparameter α? How is this hyperparameter determined?

**Ethical Concerns:**

["NO or VERY MINOR ethics concerns only"]

**Final Justification:**

One of the key challenges in sample-efficient reinforcement learning is achieving strong performance across diverse modalities and environments without relying on large computational budgets. This paper presents a solution that combines established techniques in a novel and practical way, demonstrating state-of-the-art results across multiple benchmarks. While the contribution is primarily empirical, the thorough ablations and observed synergies offer valuable insights to the community. Given its practical relevance and empirical strength, I believe the paper should be considered above the acceptance threshold.

**Limitations:**

yes

**Quality:**

3

**Strengths And Weaknesses:**

Strength:

- The author has presented compelling results that demonstrate the state of the art. Notably, on Atari 100K, Simulus achieved a human-normalized quartile mean score of 0.99, significantly outperforming prior works. The proposed method also achieves competitive performance on continuous control and multimodal benchmarks.

- The ablation study is well-conducted and provides clear insights into the contribution of each component. The results convincingly show that the final performance is the product of synergistic effects, not a single new trick. The discovery that intrinsic rewards are effective in reducing model uncertainty even in reward-rich environments is a valuable insight for the community.

- This paper is well written, clearly structured, and easy to understand.  Also, the code and model weights are open-sourced, which enables the paper being able to be reproduced.

Weakness:

- The computational cost of the model will be a problem.  Just as the author said, "token-based world model agents remain significantly slower to train than other baselines in sample-efficient RL."

- While this paper presents a system that demonstrates good performance and is empirically validated, its core contribution lies in the combined application of existing techniques within the TBWM framework. These components have been considered separately in prior work, and this work does not introduce entirely new learning objectives or architectural mechanisms. Thus, the novelty of this paper is systematic and empirical, rather than algorithmic.

---

> ### Author Rebuttal · Authors · 2025-07-30
>
> We thank reviewer nS4G for their time and effort in reviewing our paper, for their constructive and valuable feedback, and for the positive review.
>
> We appreciate that reviewer nS4G acknowledged many of the strengths of our work, including the significant improvement in IQM over prior works (Atari 100K) and overall strong performance, the quality of our ablations study, and the importance of our results and insights to the community.
>
>
> #### Weakness:
>
> `W1`:
> > The computational cost of the model will be a problem. Just as the author said, "token-based world model agents remain significantly slower to train than other baselines in sample-efficient RL."
>
> `R1`:
> While current token-based methods are indeed slower to ***train*** than some baselines (notably DreamerV3 and its variants), the overall training cost of Simulus remains reasonable. For example, DIAMOND, a recent baseline, requires nearly 3 days of training per Atari game on an RTX 4090 GPU, whereas Simulus completes training in approximately 11.9 hours under the same hardware. We observe that recent baselines generally incur higher compute costs, and we thank reviewer nS4G for prompting us to clarify this point.
>
> In addition, we argue that this should not be seen as a major drawback of our work, for several reasons:
> 1. Most importantly, architectural inefficiency of TBWMs does not diminish our main findings: our ablations show that combining intrinsic motivation, prioritized world-model replay, and regression-as-classification substantially improves the performance of world model agents in sample-efficiency settings, even in dense-reward environments. These insights are valuable regardless of training efficiency.
> 2. Our implementation builds on REM with a naive RetNet, unlike the highly optimized RNN/CNN/Transformer baselines. There is substantial headroom for improving efficiency in TBWMs, though this lies outside the scope of this work.
> 3. While training may be slower, the resulting controller is lightweight and decoupled from the world model, enabling fast and cheap inference. This offers a practical advantage over methods like Dreamer, which require the full model at test time.
>
> `W2`:
> > While this paper presents a system that demonstrates good performance and is empirically validated, its core contribution lies in the combined application of existing techniques within the TBWM framework. These components have been considered separately in prior work, and this work does not introduce entirely new learning objectives or architectural mechanisms. Thus, the novelty of this paper is systematic and empirical, rather than algorithmic.
>
> `R2`: While new learning objectives or architectural mechanisms are typical indicators of progress, demonstrating novel insights using combinations of existing methods (e.g., that combining intrinsic motivation, prioritized world-model replay, and RaC significantly boosts the performance of world model agents in sample-efficiency settings, even in dense-reward environments) can be equally valuable and impactful, as such combinations often require significant effort and insight.
> A notable example is the influential Rainbow algorithm [1] (3238 citations), which integrates known techniques to achieve strong performance. Rainbow gained significant influence despite combining existing techniques and lacking algorithmic novelty.
> We believe that our work belongs to the category of "Rainbow-style" empirical contributions, and that the absence of new algorithmic mechanisms should not be considered a weakness, as the paper offers important empirical insights instead.
>
>
> [1] Hessel, Matteo, et al. "Rainbow: Combining improvements in deep reinforcement learning." Proceedings of the AAAI conference on artificial intelligence. Vol. 32. No. 1. 2018.
>
>
> #### Questions:
>
> `Q1`:
> > Can the authors quantitatively report the computational cost, such as GPU hours or FLOPs? The observed improvement may be because of the increased computation rather than the method itself, which could be unfair to the baselines. One way to ensure fairness would be to extend the baseline models' training steps to match the total computational budget.
>
> `A1`:
> Simulus training times were reported in Appendix C (P. 26).
> In addition, we measured the run time of each component of Simulus, in each benchmark, using a RTX 4090 GPU, and also included a baseline where the intrinsic rewards and prioritized replay are disabled (No IR + No PR) to demonstrate the negligible overhead of these components.
>
> |                     | Simulus - Atari (No IR+No PR) | Simulus - Atari | Simulus - DMC | Simulus - Craftax |
> |---------------------|------|---------|---|---|
> |**Tokenizer**        |         |   |   |
> |    Step (ms)        | 73.0 | 73.0 | N/A  | N/A  |
> |    Epoch (sec)      | 14.6 (200 steps) | 14.6 (200 steps)  | N/A  | N/A  |
> |**World Model**      |      |         |   |   |
> |    Step (ms)        | 149  | 151.2   | 54  | 76  |
> |    Epoch (sec)      | 29.8 (200 steps)     | 30.2  (200 steps) | 16.2 (300 steps)  | 7.6 (100 steps)  |
> |**Controller**       |      |         |   |   |
> |    Step (ms)        | 360     | 380     | 254  | 442  |
> |    Epoch (sec)      | 28.8 (80 steps)   | 30.4 (80 steps)   | 25.4 (100 steps)  | 22.1 (50 steps)  |
> | ***Total***         | ~11.6 hrs  | ~11.9 hrs    | ~11.3 hrs  | ~3.4 days  |
> |  (sec)              | = 14.6 x 595 + 29.8 x 575 + 28.8 x 550     | = 14.6 x 595 + 30.2 x 575 + 30.4 x 550  |= 16.2 x 985 + 25.4 x 980   |= 7.6 x 9750 + 22.1 x 9700 |
>
> Please note that some measurements differ slightly (shorter) from the reported values, possibly due to updated drivers.
> We will add the above runtime analysis to the appendix of the next revision of the paper.
>
> To address the fairness concern, we note that **our ablations clearly demonstrate that the observed improvements stem from the proposed components and their synergy**. Disabling these modules leads to a significant performance drop, *despite using the same compute and model size across all ablations*.
>
> For completeness, we also consider each of the benchmarks:
> 1. Atari 100K: the run times, model size (~35M), and compute of Simulus are standard and similar to those of the other baselines.
> 2. DMC: The only available baseline is DreamerV3, which used a 12M parameters model, and a replay ratio of 512. The replay ratio is the number of steps the model is trained on per collected environment step. In comparison, on DMC Simulus uses ~8.3M parameters and a replay ratio of 128 and 96 for the controller and world model, respectively. Hence, there are no fairness issues on DMC as well.
> 3. Craftax: As Craftax is a new benchmark (2024), the only world-model baseline we found was a closed-source concurrent work, which reports that their algorithm was trained on 8 A100 GPUs and performed ~978K world model update steps in total. In comparison, Simulus was trained on a single RTX 4090 GPU for approximately 4 days and performed ~975K world model update steps in total. Thus, if there are any fairness concerns on Craftax, they would arguably disadvantage Simulus, not favor it, as the baseline relied on substantially greater computational resources.
>
> For transparency, as a small and resource-limited lab, we conducted most of our research on a single RTX 4090 GPU. Full benchmark evaluations were performed only twice, toward the end of the project, on V100 GPUs.
>
>
>
>
> `Q2`:
> > Regarding prioritized replay, how sensitive is the model performance to the hyperparameter α? How is this hyperparameter determined?
>
> `A2`: Due to compute resources limitations (see the last paragraph in `A1` above), we could not extensively investigate the sensitivity of the hyperparameter α, unfortunately. The value we used was selected early in development based on satisfactory performance and was fixed thereafter. This stability suggests that the method is not overly sensitive to α; otherwise, it would have been unlikely to obtain strong results without tuning.

---

> > ### Comment · Reviewer_nS4G · 2025-08-06
> >
> > Thank you for the detailed rebuttal, which addresses most of my questions. The empirical results are compelling, and the ablation studies are thorough. The modular design demonstrates practical value, particularly in the thoughtful combination of established techniques. While the contributions are primarily systematic rather than algorithmic, I believe the work provides meaningful empirical insights. I encourage the authors to further elaborate on these aspects in the camera-ready version. As my initial evaluation was already positive, I will maintain my score.

---

### Official Review · Reviewer_KDoM · 2025-07-03

**Clarity:** 3
**Significance:** 4
**Originality:** 3
**Rating:** 4
**Confidence:** 5

**Summary:**

In this paper authors deals with the token based world models and by using them they aim to improve significantly on the sample efficiency. This paper has four contributions: possibility to use multimodal inputs, curiosity type idea for intrinsic reward, prioritized replay buffer and the use of regression in reward modeling.

**Questions:**

- Have I undersrtood correctly that world model algorithms are typically considered to be on-policy? So then could you explain how prioritized replay buffer is used?
- In my lab we have ran a multiple WM type experiments with atari benchmarks and we have found as are also found in Dreamer set of papers that Atari is actually sometimes quite hard. Reason is that in WM you represent state by a latent code and some Atari games have really small details. Especially hard is Breakout where many times difference in successive states is ball that is only one pixel wide. That details is very easily lost in WN type latent modeling. What about your proposed method? I see some examples in the appendices, but images are too small to see what is happening.

**Ethical Concerns:**

["NO or VERY MINOR ethics concerns only"]

**Final Justification:**

Understandably authors were not able to perform large-scale experiments. However, authors well explained why DreamerV3 was not used in the craftax experiments. I had overlooked that it was actually multimodal input experiment and not vision experiment, which is was experimented in a separate Atari benchmark.

**Limitations:**

Yes

**Paper Formatting Concerns:**

-

**Quality:**

3

**Strengths And Weaknesses:**

Strengths:
- The proposed WM obtains significant sample-efficiency in the as shown in Fig 5 (craftax)
- WM is based on the REM paper and significantly improves on that is shown in the ablation experiments. In these experiments authors show that the modifications they invented were truly missing in the original REM.

Weaknesses:
- Experiments are a bit suspect. Different environments are experimented with a different set of algorithms. Especially problematic is that for craftax no Dreamer v3 is used! Noting that the original Crafter environment is _designed_ for Dreamer v2 and the experiments shown in the Crrafter github repo author shows that Dreamer v2 clearly wins over any other algorithm.
- As far as I see authors have capped their intection budget ot 100k environment interactions and that itself is ok. It is ok to focus in sample-efficiency regime. But it would be also really good to see what happens model is let to use millions of steps. This kind of experiment(s) would give full disclosure to the reader.

---

> ### Author Rebuttal · Authors · 2025-07-30
>
> We thank reviewer KDoM for their time and effort in reviewing our paper and for their constructive and valuable feedback.
>
> We appreciate that reviewer KDoM recognizes the significant improvement in sample-efficiency over REM and other baselines, and the quality of our ablations study.
>
> #### Weaknesses:
>
> `W1.1`:
> > Experiments are a bit suspect. Different environments are experimented with a different set of algorithms.
>
> `W1.2`:
> > Especially problematic is that for craftax no Dreamer v3 is used! Noting that the original Crafter environment is designed for Dreamer v2 and the experiments shown in the Crrafter github repo author shows that Dreamer v2 clearly wins over any other algorithm.
>
> `R1.1`: The reason for the differences in baselines between environments stems from the fact that except for DreamerV3, no other world model *baseline* is designed to handle multiple modalities. For example, the DIAMOND, STORM, and TWM baselines are all designed for image inputs and discrete action spaces. Hence, these baseline can not be evaluated on the other benchmarks.
>
> The baselines we used for Craftax include one world model approach which was design specifically for Craftax, and a few other simpler baselines from the original Craftax paper which we included as additional reference points for the sake of completeness.
>
> `R1.2`: While we would have liked to include DreamerV3 as an additional baseline for the Craftax benchmark, there are two limitations that prevented us from doing so:
>
> First, as a small and resource-limited lab, we conducted most of our research on a single RTX 4090 GPU. Full benchmark evaluations were performed only twice, toward the end of the project, on V100 GPUs. As such, we prioritized benchmarks with publicly available baseline results, since we could not afford to evaluate other baselines ourselves.
>
> Second, unlike *Craftax*, observations in *Crafter* are images. Hence, we preferred Craftax as (1) it offers both multi-modality and a native token modality which is absent from our other benchmarks, and (2) we already included an image-based benchmark (Atari 100K). While DreamerV2 and V3 have demonstrated strong performance on Crafter, the other baselines used for comparison are primarily model-free methods (e.g., PPO), which are known to be sample-inefficient. Since results for competing world model agents such as STORM or IRIS are not available, it remains unclear whether the Dreamer algorithms are truly superior on Crafter.
>
> Overall, we have made an honest effort to include as many relevant and recent baselines in each of our results as possible within our limitations.
>
>
>
>
> `W2`:
> > As far as I see authors have capped their intection budget ot 100k environment interactions and that itself is ok. It is ok to focus in sample-efficiency regime. But it would be also really good to see what happens model is let to use millions of steps. This kind of experiment(s) would give full disclosure to the reader.
>
> `R2`: While we agree that large-scale experiments with significantly more data and compute would make for an interesting study, such experiments are unfortunately beyond our reach due to the strict compute limitations outlined in `R1.2`. It is worth noting that the interaction budgets in DMC and Craftax are 500K and 1M steps, respectively.
>
> That said, we believe the core insights of our work remain broadly applicable and valuable, even in the absence of large-scale validation.
>
>
>
>
> #### Questions:
> `Q1`:
> > Have I undersrtood correctly that world model algorithms are typically considered to be on-policy? So then could you explain how prioritized replay buffer is used?
>
> `A1`: Yes, current world model agents typically rely on on-policy reinforcement learning. While “prioritized replay” usually refers to sampling experience based on TD-error for RL updates, in our case it applies to the world model: we prioritize *trajectory segments* based on their *world model prediction error*. Concretely, during world model training, part of each batch consists of segments sampled according to this error.
>
>
>
>
> `Q2`:
> > In my lab we have ran a multiple WM type experiments with atari benchmarks and we have found as are also found in Dreamer set of papers that Atari is actually sometimes quite hard. Reason is that in WM you represent state by a latent code and some Atari games have really small details. Especially hard is Breakout where many times difference in successive states is ball that is only one pixel wide. That details is very easily lost in WN type latent modeling. What about your proposed method? I see some examples in the appendices, but images are too small to see what is happening.
>
> `A2`: The observations raised by reviewer KDoM are highly relatable. We have encountered similar challenges and devoted significant effort to addressing them. Based on our experience,the following strategies have proven effective in mitigating these issues, especially in low-data settings:
> 1. A modular design effectively mitigates optimization interference issues similar to those observed in RSSM-based methods (e.g., Dreamer), as demonstrated in Appendix B.1. Empirically, we found that such interference often leads to "blurred" reconstructions, an effect also noted in prior work, which exacerbates the difficulty of preserving fine details in the learned representations.
> 2. Unlike continuous auto-encoders, which offer limited control over the complexity of learned representations, discrete methods allow explicit control via the vocabulary size. We found that restricting the vocabulary is particularly beneficial in low-data regimes. This insight extends beyond image observations, for example, we observed similar benefits in DMC environments. Moreover, smaller vocabularies affect not only the reconstruction quality but more importantly ***simplify the dynamics modeling task***, i.e., how difficult it is for the world model to learn and predict transitions.
> 3. We also conjecture that discrete methods enhance the separability of learned representations. This may be due to the inherently fixed and well-separated latent codes (e.g., in FSQ [1]) or due to regularization mechanisms in methods like VQVAE. Such separability can further aid the model in preserving fine details, such as those found in Atari games like Breakout.
>
> We thank reviewer KDoM for raising this question. As these insights could be valuable to the community, we propose to include them in the appendix of the next revision of the paper.
>
>
>
> [1] Mentzer, F., Minnen, D., Agustsson, E., & Tschannen, M. (2023). Finite scalar quantization: Vq-vae made simple. arXiv preprint arXiv:2309.15505.

---

> > ### Comment · Reviewer_KDoM · 2025-08-02
> >
> > I thank authors for their diligent rebuttal to my points in the weaknesses Section and answers to my questions.
> >
> > About R1.1 and R1.2, I totally understand the compute budget limitations. I understand the point about multimodal inputs in craftax and that being the point about craftax experiments. However, it would be really good if for the next revision Dreamer v3 results could be added, if for no other reason but for the completeness sake.
> >
> > About R2, our experience shows that large interaction budgets can reveal surprising phenomena that are not available in the low interaction budget regimes. You can also see that in the Dreamer papers, if my memory serves, authors used quite a large interaction budgets to able to obtain Diamond in the Minecraft experiment. Here again, I would like authors to add larger scale experiments to the next revision of their paper.
> >
> > I am quite satisfied with the rebuttal and I am willing to raise my score.

---

> > > ### Author Response · Authors · 2025-08-03
> > >
> > > We sincerely thank the reviewer for their thoughtful engagement with our rebuttal and for the willingness to reconsider their score. We are grateful for the recognition of our responses to the points raised in the weaknesses section and the understanding regarding our compute budget limitations.
> > >
> > > We agree that including DreamerV3 would improve the completeness of our evaluation. While training DreamerV3 under our current constraints will be challenging, we are committed to include it as a baseline in the next revision of the paper.
> > >
> > > Regarding the request for large-scale experiments with extended interaction budgets, we would like to offer the following context. At present, we only have access to V100 GPUs. Running a single Atari experiment with 100K interaction steps takes approximately 29 hours on a V100, meaning that scaling to 10M steps would increase the runtime by a factor of 100, i.e., 2900 hours (or ~120 days) for just a single seed. Standard evaluation protocols typically require at least 5 random seeds per environment to ensure statistical reliability. Similar constraints apply to our other benchmarks. Unfortunately, such a study is far beyond our available resources, even for a small subset of tasks.
> > > For comparison, the DreamerV3 paper, which reported the large scale experiments mentioned, was produced by researchers at Google DeepMind, one of the most well resourced labs in the field.
> > >
> > > While exploring such surprising phenomena can yield excellent insights and important outcomes, this is largely orthogonal to the contributions of our paper. Specifically, the insight that intrinsic motivation (when combined with the other components) can significantly improve performance in low data settings, even in reward-rich environments, is valuable and important. However, as the interaction budget scales, we expect that this insight will become less surprising, as the cost of model-oriented exploration decreases. Hence, we view large-scale validation as a promising direction for future work that complements our current contributions.
> > >
> > > We thank the reviewer again for their constructive feedback and support.

---

### Official Review · Reviewer_cNHU · 2025-07-23

**Clarity:** 3
**Significance:** 2
**Originality:** 1
**Rating:** 2
**Confidence:** 5

**Summary:**

The paper presents Simulus, a token‑based world model (TBWM) agent aimed at improving sample efficiency in RL. The authors propose a modular pipeline—separating representation learning, the world model, and the controller—and weave in four key components: (1) multi‑modal tokenization to handle heterogeneous observation/action spaces, (2) intrinsic motivation targeted at reducing epistemic uncertainty, (3) prioritized replay within the world model, and (4) a regression‑as‑classification scheme for reward/return prediction. Empirically, Simulus attains state‑of‑the‑art results across several benchmarks (e.g., Atari 100K, DMC, Craftax-1M).

**Questions:**

A few questions that the authors can please elaborate:

1. Can you quantify the wall‑clock training cost of Simulus relative to key baselines (e.g., GPU hours per environment) and identify which modules dominate that cost?

2. How sensitive is performance to the tokenization granularity for continuous vectors, and have you explored strategies to cap sequence length without hurting returns?

3. Given the scarcity of multi‑modal RL benchmarks, do you have plans (or preliminary results) for additional domains beyond Craftax that stress different modality combinations?

4. Where does Simulus underperform or fail to scale (e.g., very high‑dimensional observations or extremely long horizons), and what do you see as the most promising path to reduce the compute burden?

**Ethical Concerns:**

["NO or VERY MINOR ethics concerns only"]

**Final Justification:**

The reason for my final score is that despite its solid engineering and strong empirical results, the paper does not present a genuinely novel contribution, as it combines four established techniques; multi-modal tokenization, intrinsic-motivation rewards, prioritized world-model replay, and regression-as-classification, in Token Based World Models without offering a unifying theoretical framework or evaluating why the combinations of these techniques actually result in improved performance (e.g., see Section 3.3 Ablation, lines 295-297). Whatever explanation is presented feels more like conjecture and after-thoughts rather than a well-reasoned and theorteically grounded insight. The paper's key design choices such as vocabulary sizes and return binning remain heuristic, while the limited multi-modal validation in a single environment (Craftax) fails to demonstrate true multi-modality, leaving the work feeling like an incremental compilation of known methods rather than a coherent advance. While this is a thorough work in itself, I am not sure if this constitutes a signficant enough contribution to be presented at NeurIPS. Therefore, I recommend a rejection of this paper.

**Limitations:**

Limitations have been discussed briefly in Section 5, which are sufficient.

**Paper Formatting Concerns:**

None, the paper is excellently formatted and very detailed. Top-notch work.

**Quality:**

2

**Strengths And Weaknesses:**

Strenghts:

For this reviewer, Simulus’s main strength is its modular setup: the representation, world model, and controller can each be built and tuned on their own. The paper also stitches together several underused ideas—multi‑modal tokenization, intrinsic rewards to cut down epistemic uncertainty, prioritized world‑model replay, and regression‑as‑classification—showing they work better together than alone for sample efficiency.

Additionally, on the results side, Simulus posts state‑of‑the‑art sample efficiency for planning‑free world models on Atari 100K, the DeepMind Control Suite, and Craftax‑1M. It hits human or even superhuman scores on multiple Atari titles and reliably matches or beats baselines in both continuous and multi‑modal settings.

The framework pushes TBWMs past the usual vision + discrete‑action niche. Thanks to the modular multi‑modal tokenization, it handles continuous control (DMC) and richer multi‑modal environments (Craftax), which nudges TBWMs closer to real‑world RL problems.

Another important and useful contribution of the paper is that the ablations are actually useful: intrinsic rewards, prioritized replay, and RaC return prediction are each tested in isolation. Intrinsic rewards, in particular, come out as a major contributor, and the combined effect of all pieces is clear. Moreover, the paper spells out methods, architectures, hyperparameters, and experimental details; code and pretrained weights are available and will be made public.

Some of the things I find to be on the weaker side are:

1. Training is expensive: tokenizing continuous vectors blows up sequence length, and the authors admit TBWMs (including Simulus) train much slower than standard baselines. Even if modularity helps inference, the upfront cost is a real barrier for groups without large compute.

2. Multi‑modality isn’t fully stress‑tested: the lack of rich multi‑modal RL benchmarks limits how broadly they can validate the approach. Craftax is a start, but more varied tasks would make the claim stronger.

3. While the appendix discusses interfering objectives in RSSM optimization and provides preliminary results on PWM vs. PWM-decoupled, this section feels somewhat tangential to the core contributions of Simulus itself. While interesting, it could be refined to more directly link how Simulus's design avoids or mitigates such interference, rather than just observing it in a related model.

---

> ### Author Rebuttal · Authors · 2025-07-30
>
> We thank reviewer cNHU for their time and effort in reviewing our paper, for their constructive and valuable feedback, and for the positive review.
>
> We appreciate the reviewer’s recognition of Simulus’s modular design, the effective combination of underused techniques, and its state-of-the-art sample efficiency across diverse benchmarks. They also valued our extension of TBWMs beyond vision-only tasks, the clarity and insight of our ablations, especially on intrinsic rewards, and our commitment to transparency through detailed documentation and open-sourcing.
>
>
>
> #### Weaknesses:
> `W1`:
> > Training is expensive: tokenizing continuous vectors blows up sequence length, and the authors admit TBWMs (including Simulus) train much slower than standard baselines. Even if modularity helps inference, the upfront cost is a real barrier for groups without large compute.
>
>
> `R1`: While the training cost of current TBWMs is indeed higher than some baselines (notably DreamerV3 and its variants), it remains comparable to or lower than others. For instance, the DIAMOND baseline requires nearly 3 days of training per Atari game on an RTX 4090 GPU, compared to ~11.9 hours for Simulus. Moreover, as relatively recent methods, TBWMs still offer considerable room for optimization, as we also discuss below (`A4`).
>
> Crucially, we believe that the architectural inefficiency of current TBWMs should not detract from the core insights of our work. Our ablations demonstrate that combining intrinsic motivation, prioritized world-model replay, and regression-as-classification significantly boosts the performance of world model agents in sample-efficient settings, even in dense-reward environments. These findings are meaningful regardless of implementation efficiency.
>
> For transparency, as a small and resource-limited lab, we conducted most of our research on a single RTX 4090 GPU. Full benchmark evaluations were performed only twice, toward the end of the project, on V100 GPUs.
>
>
>
> `W2`:
> > Multi‑modality isn’t fully stress‑tested: the lack of rich multi‑modal RL benchmarks limits how broadly they can validate the approach. Craftax is a start, but more varied tasks would make the claim stronger.
>
>
> `R2`:
> We agree that extensive evaluation on rich multi-modal environments remains a limitation, as we also acknowledged in our limitations section. However, our notion of a "multi-modality tokenization framework" also encompasses the ability to handle diverse *uni-modal* environments. Under our three-benchmark budget, already more comprehensive than the 1–2 benchmarks typical in this literature, we covered a wide range of modalities.
> Given that standard and well-established sample-efficiency benchmarks like Atari 100K and DMC are uni-modal, and since such environments are far more prevalent, demonstrating strong performance across them is arguably more important, and also provides concrete support for the framework’s multi-modality capabilities.
>
> Therefore, while we recognize the limitation, we believe the lack of additional multi-modal tasks should not be considered a major concern.
>
>
>
> `W3`:
> > While the appendix discusses interfering objectives in RSSM optimization and provides preliminary results on PWM vs. PWM-decoupled, this section feels somewhat tangential to the core contributions of Simulus itself. While interesting, it could be refined to more directly link how Simulus's design avoids or mitigates such interference, rather than just observing it in a related model.
>
>
> `R3`:
> These results were included to serve two main purposes:
> 1. To support the claim made in the introduction that separate optimization objectives reduce interference, by first demonstrating that such interference exists, and then showing that decoupling the objectives effectively mitigates it.
> 2. To motivate the modular design of Simulus.
>
> We assume this behavior holds independently of the underlying architecture (RSSM in this case), as the interference arises from coupled optimization rather than architectural specifics. This directly supports the idea that Simulus’s modular design helps avoid similar optimization issues. We thank reviewer cNHU for this thoughtful observation. In the next revision, we will make the connection between these results and Simulus’s design more explicit to improve clarity.
>
>
> #### Questions:
>
> `Q1`:
> > Can you quantify the wall‑clock training cost of Simulus relative to key baselines (e.g., GPU hours per environment) and identify which modules dominate that cost?
>
>
> `A1`:
> Simulus training times were reported in Appendix C (P. 26).
> In addition, we measured the run time of each component of Simulus, in each benchmark, using a RTX 4090 GPU, and also included a baseline where the intrinsic rewards and prioritized replay are disabled (No IR + No PR) to demonstrate the negligible overhead of these components.
>
> |                     | Simulus - Atari (No IR+No PR) | Simulus - Atari | Simulus - DMC | Simulus - Craftax |
> |---------------------|------|---------|---|---|
> |**Tokenizer**        |         |   |   |
> |    Step (ms)        | 73.0 | 73.0 | N/A  | N/A  |
> |    Epoch (sec)      | 14.6 (200 steps) | 14.6 (200 steps)  | N/A  | N/A  |
> |**World Model**      |      |         |   |   |
> |    Step (ms)        | 149  | 151.2   | 54  | 76  |
> |    Epoch (sec)      | 29.8 (200 steps)     | 30.2  (200 steps) | 16.2 (300 steps)  | 7.6 (100 steps)  |
> |**Controller**       |      |         |   |   |
> |    Step (ms)        | 360     | 380     | 254  | 442  |
> |    Epoch (sec)      | 28.8 (80 steps)   | 30.4 (80 steps)   | 25.4 (100 steps)  | 22.1 (50 steps)  |
> | ***Total***         | ~11.6 hrs  | ~11.9 hrs    | ~11.3 hrs  | ~3.4 days  |
> |  (sec)              | = 14.6 x 595 + 29.8 x 575 + 28.8 x 550     | = 14.6 x 595 + 30.2 x 575 + 30.4 x 550  |= 16.2 x 985 + 25.4 x 980   |= 7.6 x 9750 + 22.1 x 9700 |
>
> Please note that some measurements differ slightly (shorter) from the reported values, possibly due to updated drivers.
> For comparison, on Atari, REM runs for ~ 11 hrs, and on DMC, DreamerV3 runs for ~7.2 hrs on a A100 GPU where Simulus would run for ~20 hrs on the same GPU (assuming that V100 is twice slower than A100). Note that Simulus is not highly optimized, and as stated in the limitations, vector tokenization is highly inefficient as it leads to excessive sequence lengths.
> On Craftax, the only world model baseline we found (concurrent work) uses 8 A100 GPUs for training. Hence, it is not directly comparable, as such setup requires significantly more memory.
>
>
>
> `Q2`:
> > How sensitive is performance to the tokenization granularity for continuous vectors, and have you explored strategies to cap sequence length without hurting returns?
>
>
> `A2`: Discretizing continuous actions does not inherently limit controller precision, as this can be compensated by a shorter control interval or higher control frequency. However, there is a clear trade-off: large action spaces allow lower decision frequency but increase exploration cost, while smaller action spaces require more frequent decisions, increasing computational load. In our experiments, we did not encounter any issues related to action discretization.
>
> A key insight in the context of world models is that the size of the vocabulary controls not only reconstruction quality but also the ***complexity of the dynamics modeling task***, i.e., how difficult it is for the world model to learn and predict transitions. We found that using smaller vocabularies significantly improves sample efficiency, not only in continuous control but more broadly. While this may seem intuitive, we believe it is worth explicitly including in the appendix of our next revision, especially as it was also highlighted by other reviewers. We thank reviewer cNHU for raising this point.
>
> Regarding the second part of the question, we did not explore strategies to cap sequence length, as this would require substantial additional effort and is largely orthogonal to our paper’s main contributions. Our insights are independent of architectural (in)efficiencies, and we leave this line of work to future research. That said, we propose a simple direction in `A4` below.
>
>
> `Q3`:
> > Given the scarcity of multi‑modal RL benchmarks, do you have plans (or preliminary results) for additional domains beyond Craftax that stress different modality combinations?
>
>
> `A3`:
> While we made sincere efforts to devise creative solutions to this challenge, we were ultimately unable to identify an adequate benchmark or evaluation protocol. One preliminary idea involved constructing a multi-modal environment by running two uni-modal environments in parallel. However, we found this approach difficult to justify, as it does not reflect real-world settings and can be effectively handled using separate uni-modal controllers.
>
> We hope future benchmarks will build on modern video games where audio conveys crucial information. In these scenarios, agents that can leverage such cues gain a clear advantage.
>
>
> `Q4`:
> > Where does Simulus underperform or fail to scale (e.g., very high‑dimensional observations or extremely long horizons), and what do you see as the most promising path to reduce the compute burden?
>
>
> `A4`: While we have not explored large scale training due to compute limitations, we observed that in environments with continuous (vector) observations the scaling would be particularly inefficient, as mentioned in our limitations section, due to the inefficient tokenization that leads to excessive sequence lengths.
>
> We believe this compute burden can be significantly reduced with improved design. In particular, architectural changes that better balance intra-token and inter-token information, e.g., by aggregating fixed-size groups of latents (akin to “patching”), offer a simple and promising direction. Furthermore, our implementation uses a naive RetNet variant, and we expect substantial efficiency gains from sequence model optimization.

---

> > ### Comment · Reviewer_cNHU · 2025-08-06
> >
> > I thank the authors for their detailed reply and engaging in the rebuttal discussion.
> >
> > After reading other reviews, I have doubts about the methodological contribution of this work. While it is a substantial empirical study, it combines existing approaches and hence limits the originality of the work. While the authors claim that their work be viewed in the same line as Rainbow (and perhaps may I add more recently the Bigger Better Faster (BBF) paper), which combine existing techniques to achieve SOTA, questions still remain about the novelty and signifiance of the contributions. The empirical rigor and engineering work is commendable, but does it alone merit acceptance is a difficult question to answer.
> >
> > I will combine the opinions of other reviewers (and their original scores) and give a final score.

---

> > > ### Author Response · Authors · 2025-08-08
> > >
> > > We thank reviewer cNHU again for their thoughtful review and for acknowledging the empirical rigor and engineering contributions.
> > >
> > > > The empirical rigor and engineering work is commendable, but does it ***alone*** merit acceptance is a difficult question to answer.
> > >
> > > We emphasize that our work offers contributions that go beyond empirical rigor and engineering effort. In addition to demonstrating state-of-the-art performance across diverse benchmarks, a central contribution is the insight that combining intrinsic motivation, prioritized *world model* replay, and regression-as-classification leads to substantial improvements in sample efficiency, even in environments with dense rewards and despite the high interaction cost of model-oriented exploration. We believe this insight is valuable to the community and may become standard practice in future practical world model agents.
> > >
> > > While the reviewer's concerns focus on the fact that these are not novel components, we highlight that the contribution lies in the discovery that their combination yields significant gains in settings where intuitively intrinsic motivation should be less appropriate (sample-efficiency and/or dense-reward settings).
> > >
> > > In comparison to recent world model works [1][2][3] published at top conferences, we believe our work meets a similarly high standard in both insight and empirical contribution, particularly in addressing challenges in sample-efficient reinforcement learning.
> > > We therefore hope our work will be judged by the same standards and not more harshly.
> > >
> > >
> > >
> > > [1] Zhang, W., Wang, G., Sun, J., Yuan, Y., & Huang, G. (2023). Storm: Efficient stochastic transformer based world models for reinforcement learning. Advances in Neural Information Processing Systems, 36, 27147-27166.
> > >
> > > [2] Robine, J., Höftmann, M., Uelwer, T., & Harmeling, S. Transformer-based World Models Are Happy With 100k Interactions. In The Eleventh International Conference on Learning Representations.
> > >
> > > [3] Micheli, V., Alonso, E., & Fleuret, F. Efficient World Models with Context-Aware Tokenization. In Forty-first International Conference on Machine Learning.

---

### Decision · Program_Chairs · 2025-09-17

**Decision:**

Reject

**Comment:**

In this submission, the authors introduce a novel method called Simulus - a token-based world model which can be used to improve the sample efficiency of reinforcement learning (RL) agents. Simulus is composed of 4 main components, namely the multimodal tokenization, intrinsic motivation for uncertainty-driven exploration, prioritised world model regeneration, and regression classification for reward prediction. In their experiments, the authors show that Simulus can attain state-of-the-art performance (without planning) in Atari 100K, DeepMind Control Suite, and Craftax-1M benchmarks.

The reviewers have noted several strengths of the paper. They highlight its significant sample-efficiency and compelling empirical results. The reviewers also agree that the manuscript is well-written and easy to follow. This said, several weakness have also been noted. Several reviewers highlighted limited methodological novelty in the manuscript, with the main improvements coming from integrating existing, well-established approaches. Another issue raised was the limited number of experiments with multi-modal environments. During the rebuttal, the authors have address some of the concerns raised by the reviewers. Nonetheless, some reservations still remained and no reviewer was willing to champion the paper in its current form.

I believe that the submission has a lot of potential. I'd strongly recommend that the authors address the reviewers' comments thoroughly before re-submitting.